# Pacemaking function of two simplified cell models

**Maxim Ryzhii**[1]*, **Elena Ryzhii**[2]

**1** Complex Systems Modeling Laboratory, University of Aizu, Aizu-Wakamatsu, Japan, **2** Department of Anatomy and Histology, Fukushima Medical University, Fukushima, Japan

* m-ryzhii@u-aizu.ac.jp

## Abstract

Simplified nonlinear models of biological cells are widely used in computational electrophysiology. The models reproduce qualitatively many of the characteristics of various organs, such as the heart, brain, and intestine. In contrast to complex cellular ion-channel models, the simplified models usually contain a small number of variables and parameters, which facilitates nonlinear analysis and reduces computational load. In this paper, we consider pacemaking variants of the Aliev-Panfilov and Corrado two-variable excitable cell models. We conducted a numerical simulation study of these models and investigated the main non-linear dynamic features of both isolated cells and 1D coupled pacemaker-excitable systems. Simulations of the 2D sinoatrial node and 3D intestine tissue as application examples of combined pacemaker-excitable systems demonstrated results similar to obtained previously. The uniform formulation for the conventional excitable cell models and proposed pacemaker models allows a convenient and easy implementation for the construction of personalized physiological models, inverse tissue modeling, and development of real-time simulation systems for various organs that contain both pacemaker and excitable cells.

## Introduction

Nowadays computer modeling of various organs and tissues is an indispensable part of physiology research. Computational models of different levels of complexity are being utilized, with complex ion-channel multi-variable models on the top of the list. A rigorous review of the cardiac models was presented in [1]. The number of such models and their updates increases yearly following new physiological findings and measurements. Consisting of many differential equations for ion channels and their gate formulations, the models provide a detailed description of cell behavior under various normal and pathological conditions, including, for example, the influence of drugs [2]. The number of variables and parameters in such models can reach several dozens, leading to the necessity to utilize significant computational resources, even despite current progress in and availability of graphical processing units (GPU) and multicore processors [3]. Moreover, these precise ion-channel models, in particular, cardiac ones, usually require small enough time steps and mesh sizes to provide calculation stability in the case of tissue simulation [2, 4], leading to hours and even days of calculation using normal

**Data Availability Statement:** All relevant data are within the manuscript and its Supporting information files.

**Funding:** MR, Grant No. 20K12046, JSPS KAKENHI https://www.jsps.go.jp/english/e-grants/ The funders had no role in study design, data

collection and analysis, decision to publish, or preparation of the manuscript.

**Competing interests:** The authors have declared that no competing interests exist.

desktop computers. Thus, utilization of the biophysically based ion-channel models in real-time systems, such as equipment test-beds and systems for formal validation of medical devices [5, 6], interactive tools for computer-aided therapy planning [7], and other real-time simulation devices and platforms, is still nearly impossible.

In many cases, simplified phenomenological cell models, such as classical Van der Pol [8] (VDP), FitzHugh-Nagumo [9, 10] (FHN), and Hodgkin-Huxley [11] (HH) can be a good alternative. These models are based on a small number (usually 2—3) of variables, i.e., ordinary differential equations (ODEs). Later additions to this class include modifications of the HH and FHN models, namely Van Capelle-Durrer (VCD) [12], Aliev-Panfilov [13] (AP), Morris-Lecar [14] and its pacemaking variants [15, 16], Fenton-Karma [17], Mitchell-Schaeffer [18] (MS), and its modification by Corrado and Niederer [19] (CN). The latter two models have been used recently to simulate electrophysiology of atria, spiral wave stability, and ventricular tachycardia inducibility in patient-specific models (see review [20] and references therein). Most of the models mentioned above are included in the modeling software packages and repositories, such as openCARP [21] and Physiome Project [1, 22].

Fig 1 demonstrates the timeline of the historical development of the main simple physiological models. Solid shapes correspond to the excitable models, dashed—to the pacemaking models, and the models capable to exhibit both types of behavior are represented by both shape types. Even though most of the models shown in Fig 1 are capable to provide pacemaking operation, their utilization is limited. The VDP and FHN models and their modifications are being predominantly used as simple models of natural pacemakers in physiological simulations of different levels of complexity (see, for example, [23–28]). The VDP model of relaxation oscillator was proposed to describe general heartbeat dynamics and by its nature does not have the quiescent excitable form. The two-variable FHN model, as a reduction of the four-variable HH model of the squid giant axon action potential, was developed to model neuronal excitability. It has the properties of producing spike trains (tonic spiking) at sufficiently large stimulating constant current, and the apparent absence of a firing threshold [29], which are undesirable for simulation of other organs, such as the heart.

The disadvantage of other simplified models simulating both pacemaker and non-pacemaker action potentials of cardiac cells like VCD, Morris-Lecar, and their modifications is the significant number of parameters that limit the areas of their utilization. On the other hand, computationally lightweight models such as AP, MS, and CN are being successfully used for the solution of the cardiac inverse problem and building patient-specific models [30–33]. The MS model, however, is known to have some stability problems [19].

For multi-scale and real-time simulations, models with a modest number of parameters are preferable [28]. Such models have been implemented in many studies of electrophysiology of the heart and cardiac tissue [34–38], as well as the intestine [23, 24], stomach [39, 40], uterus [41], and bladder [42]. Recently, the AP model was used in deep learning-based reduced-order modeling [43], allowing to boost the solution of parameterized problems in cardiac electrophysiology, and in preliminary setup for hypothesis testing and verification, and model tuning before implementation of computationally expensive ion-channel models [34].

Another recently proposed type of pacemaker model developed to satisfy real-time requirements is represented by parametric [44] and resonant [45] (8–24 variables) models. They quantitatively reproduce action potential shapes and some cellular behavior but do not include several important physiological properties such as interactions due to electrotonic coupling. Although the models provide a relatively low computational load, the latter may significantly rise with the inclusion of detailed features. The above-mentioned models, however, underline the need for simple computationally efficient models, in particular, having a uniform description for pacemaking and excitable cells. These models are essential for abstracted heart models

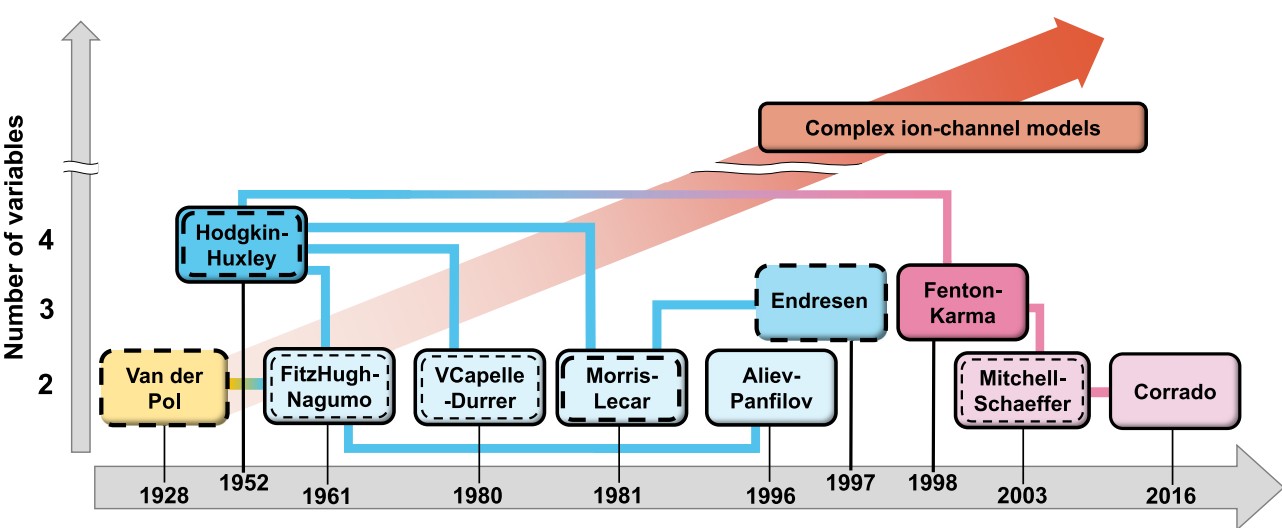

**Fig 1. Timeline of simplified physiological cell models development.** Solid shapes correspond to excitable cell models, dashed shapes—to pacemaking cell models, respectively.

with real-time simulation capabilities, where a single pacemaker cell represents a group of ion-channel pacemaker model cells [44, 46].

In this work, we consider variants of the AP and CN phenomenological models providing them intrinsic pacemaker properties (hereinafter called pAP and pCN, respectively), and demonstrate their main characteristics for both single pacemaking cells and coupled pacemaker-excitable systems.

As application examples for the proposed pacemaker models, we include simulations of the 2D cardiac sinoatrial node (SAN) model described with the pAP and AP cells, and the 3D intestinal model consisting of the pCN and CN cells.

## Methods

### Self-oscillations in the Aliev-Panfilov model

The two-variable AP model [13] proposed to describe non-oscillatory cardiac tissue that supports stable propagation of excitation waves is represented by the following set of reaction-diffusion type nonlinear ordinary differential equations:

$$\frac{\partial u}{\partial t} = c_t[ku(u-a)(1-u) - vu] + I_{ext}, \tag{1}$$

$$\frac{\partial v}{\partial t} = c_t \varepsilon[-v - ku(u-a-1)], \tag{2}$$

$$\varepsilon = \varepsilon_0 + \frac{v\mu_1}{u + \mu_2},$$

where $u$ and $v$ are normalized transmembrane potential and slow recovery variable, respectively. Parameter $k$ controlling the magnitude of transmembrane current, and parameters $\mu_1$, $\mu_2$, and $a$ are adjusted to reproduce characteristics of cardiac tissue, $\varepsilon$ sets the time scale of the recovery process, $c_t = 1/12.9$ is the time scaling coefficient introducing physical time in

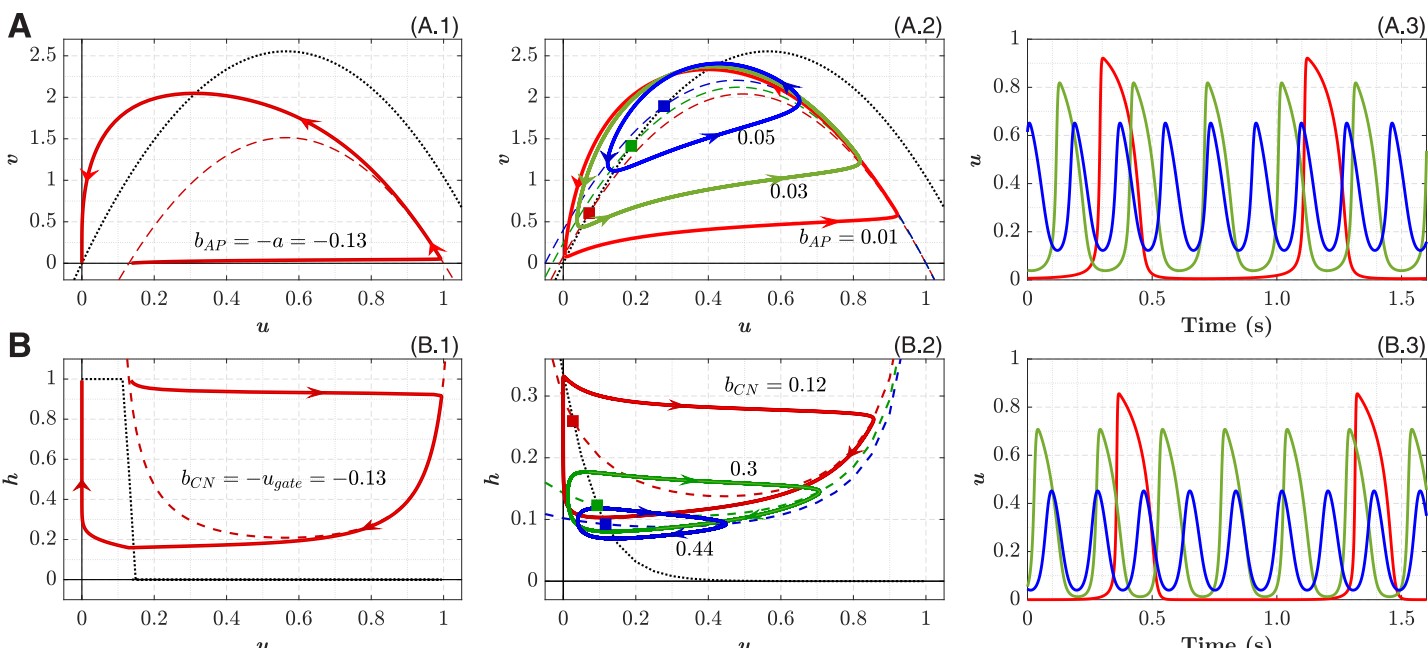

**Fig 2. Single-cell nonlinear dynamics. A. AP and pAP models**. A.1. Nullclines and phase portrait for the conventional AP model after suprathreshold stimulation. A.2. Nullclines and phase portrait for the pAP model with different bifurcation parameter $b_{AP}$. Dashed and dotted lines correspond to $u$- and $v$-nullclines, respectively. Nullcline $u = 0$ is not shown. Stable limit cycles for $b_{AP} = 0.01$, 0.03, and 0.05 are shown with corresponding EPs marked by squares of the same color. A.3. Action potentials, colors correspond to the curves in panel A.2. **B. CN and pCN models**. B.1. Nullclines and phase portrait for the conventional CN model after suprathreshold stimulation. B.2. Nullclines and phase portrait for the pCN model with different bifurcation parameter $b_{CN}$. Dashed and dotted lines correspond to $u$- and $h$-nullclines, respectively. Nullcline $u = 0$ is not shown. Stable limit cycles for $b_{CN} = 0.12$, 0.3, and 0.44 are shown with corresponding EPs marked by squares of the same color. B.3. Action potentials, colors correspond to the curves in panel B.2.

milliseconds into the system. $I_{ext} = \nabla \cdot (\mathbf{D}\nabla u)$ is external or coupling current in the case of multi-cell simulations, where $\nabla$ is a spatial gradient operator defined within the model tissue geometry, and $\mathbf{D}$ is a tensor of diffusion coefficients (in mm$^2$ms$^{-1}$) characterizing electrotonic interactions between neighboring cells via gap junctional coupling conductance.

In the conventional AP model, the left branch of the $u$-nullcline $v = k(u - a)(1 - u)$ (dashed lines in Fig 2A.1 and 2A.2) does not enter the region where $u$ is negative. The phase space trajectory (shown after suprathreshold stimulation in Fig 2A.1) is also limited in the region $u > 0$ by the second $u$-nullcline $u = 0$.

As the excitation threshold $a$ reduces, the nullcline moves up, its left branch moves toward $u < 0$. When the parameter $a$ becomes negative, the $u$-nullcline intersects the $v$-nullcline $v = -ku(u - a - 1)$ (dotted lines in Fig 2A.1 and 2A.2), creating equilibrium points (EP) in the region $u > 0$. The system of Eqs (1) and (2) undergoes Hopf bifurcation (HB), which is a typical mechanism for the onset of oscillations with a stable limit cycle (periodic orbit) in nonlinear dynamical systems [47]. For the sake of convenience, further we introduced a parameter $b_{AP} = -a$ in Eq (1), controlling the intrinsic oscillation frequency when $b_{AP} > 0$:

$$\frac{\partial u}{\partial t} = c_t[ku(u + b_{AP})(1 - u) - vu] + I_{ext}. \tag{3}$$

The idea to use the parameter $b_{AP}$ for cubic $u$-nullcline in the FHN-like systems to control the excitability range is straightforward. It has been used, for example, to study the propagation of action potential in combined pacemaking-excitable FHN model tissue [48]. However, in the

case of the AP model, which was developed primarily to represent excitable cardiac tissue, the intrinsic pacemaking function in a single cell and coupled systems was not considered yet.

The resulting phase-space geometry of the pAP model is shown in Fig 2A.2. Three stable limit cycles with $b_{AP}$ = 0.01, 0.03, and 0.05 and their EPs (marked by squares) in Fig 2A.2.

## Corrado excitable cell model and its pacemaking variant

The two-variable CN modification [19] of the ionic MS model [18] for cardiac excitable cells is represented by the following set of nonlinear differential equations:

$$\frac{\partial u}{\partial t} = \frac{hu(u - u_{gate})(1 - u)}{\tau_{in}} - \frac{(1 - h)u}{\tau_{out}} + I_{ext}, \tag{4}$$

$$\frac{\partial h}{\partial t} = \begin{cases} \dfrac{1 - h}{\tau_{open}} & \text{if } u \leq u_{gate} \\[2mm] \dfrac{-h}{\tau_{close}} & \text{if } u > u_{gate} \end{cases}. \tag{5}$$

Here $h$ is the gating variable for the inward current (sodium ion channels), $u_{gate} > 0$ is the excitation threshold potential, $\tau_{in}$, $\tau_{out}$, $\tau_{open}$, and $\tau_{close}$ are the time constants affecting the corresponding characteristic phases of the evolution of transmembrane potential $u$ (shown after suprathreshold stimulation in Fig 2B.1). As in the pAP model, the latter is also limited in the region $u > 0$ by the second $u$-nullcline $u = 0$.

To introduce pacemaking behavior into the CN model, we did the following modifications. First, the piece-wise function (Eq 5) was replaced by the formulation with the sigmoid function [11, 49] of the transmembrane potential (Eqs 7–9) with the slope factor $u_s$ (in dimensionless voltage units), similar to the approach used in [50], changing the shape of $h$-nullcline [51]. Second, we replaced $u_{gate}$ in Eq 4 with a parameter $b_{CN} = -u_{gate}$, allowing to shift independently left branch of the $u$-nullcline $h = \tau_{in}/[\tau_{out}(u - u_{gate})(1 - u) + \tau_{in}]$:

$$\frac{\partial u}{\partial t} = \frac{hu(u + b_{CN})(1 - u)}{\tau_{in}} - \frac{(1 - h)u}{\tau_{out}} + I_{ext}, \tag{6}$$

$$\frac{\partial h}{\partial t} = \frac{(h_\infty - h)}{\tau}, \tag{7}$$

$$\tau = \frac{\tau_{open}\tau_{close}}{\tau_{open} - h_\infty(\tau_{close} - \tau_{open})} \tag{8}$$

$$h_\infty = \frac{1}{2}\left[1 - \tanh\left(\frac{u - u_{gate}}{u_s}\right)\right]. \tag{9}$$

Similar to the pAP model, $b_{CN}$ becomes a parameter suitable for controlling both the CN cell excitability and the pCN intrinsic oscillation frequency (see Fig 5A.1 below).

Replacement of the Heaviside-like step function in Eq 5 for $h_\infty$ by the sigmoid voltage dependence in the limit [49] (conventional Boltzmann equation for cell membranes [11])

$$H(x) = \lim_{u_s \to 0}\left(1 + \exp\left(-\frac{x}{u_s}\right)\right)^{-1} = \frac{1}{2}\lim_{u_s \to 0}\left[1 - \tanh\left(-\frac{x}{u_s}\right)\right]$$

and the increase of the slope factor $u_s$ reduce the robustness of the CN model and enhance its propensity to spontaneous oscillations by inclination the central branch of the $h$-nullcline (dotted lines in Fig 2B.1 and 2B.2). This, together with the shift of the left branch of the $u$-nullcline (dashed lines in Fig 2B.1 and 2B.2) toward $u < 0$ region with increasing parameter $b_{CN}$, leads the system of Eqs 6 and 7 to HB, creates EP at the nullclines intersection, and provides the appearance of a stable limit cycle. The parameters $u_s$ and $u_{gate}$ together with $b_{CN}$ define the position of $u$ and $h$ nullclines intersection, and consequently the shape of phase space trajectory (see Fig 5A.1, 5B.1, and 5C.1 below).

Though the method is similar to the previously proposed pacemaking modification of the MS excitable cell model [50], the considered pCN model (Eqs 6–9) possesses different nonlinear dynamic properties.

Note the clockwise direction with respect to the second variable in the limit cycles of the pCN model (Fig 2B.1 and 2B.2) in contrast to the pAP model (Fig 2A.1 and 2A.2).

## Isolated single pacemaker cells

For single-cell cases (0D), we examined the dynamics of the pAP and pCN model cells changing various bifurcation parameters and constructing bifurcation diagrams [52]. The incremental steps of the parameters were selected individually for each model and varied depending on observed dynamics. When spontaneous oscillation appeared, we determined peak overshoot potential (POP), maximum diastolic potential (MDP), frequency, diastolic interval (DI), and action potential duration (APD) at 90% repolarization for each bifurcation parameter value upon allowing the oscillation activity to stabilize within 20–50 s.

## Coupled 1D pacemaker-excitable systems

One of the important characteristics of a pacemaking cell is its synchronization behavior under an applied load of coupled cells with a variable diffusion coefficient [53–55]. To investigate this property of the considered pacemaking cells, we set up three variants of the load—$n$ strands (cables) of matching 20 excitable cells coupled to a single pacemaker cell. Because the load in a tissue is not limited to an integer value ($n$ can be non-integer [55]), apart from the normal case ($n = 1$) we considered higher ($n = 2$) and lower ($n = 0.5$) loads.

These load-driving capabilities are essential for real-time simulation systems based on abstracted heart models, where a single pacemaker cell represents a group of ion-channel pacemaker model cells [44, 46].

**Minimal and maximal frequencies of complete 1:1 synchronization.** In a wide range of fixed values of the coupling coefficient $d = D/\Delta x^2 = 0.02$–$10.0$ ms$^{-1}$ ($D$ is the diffusion coefficient) we run multiple simulations of the pAP-AP and pCN-CN coupled systems, varying the values of the control parameters $b_{AP}$ and $b_{CN}$, respectively, calculating the resulting frequency ratio between the pacemaker cell and the 16th excitable cell in a strand (to eliminate possible effects of the boundary conditions on the last excitable cell). Next, we determined the system's lowest (minimal) and highest (maximal) frequency with complete 1:1 synchronization between the cells.

**Relationship between pacemaker cell rates and intercellular coupling.** In these simulations, we fixed the intrinsic oscillation frequency of the pacemaker cells (fixing the parameters $b_{AP}$ and $b_{CN}$) at the values close to the upper-frequency limit of 1:1 pacemaker-excitable system synchronization. Changing the intercellular coupling $d$ we examined the onset of transitions between full 1:1 and incomplete synchronizations [56], and recorded obtained frequencies of the pacemaking cell and the 16th excitable cell in the strand.

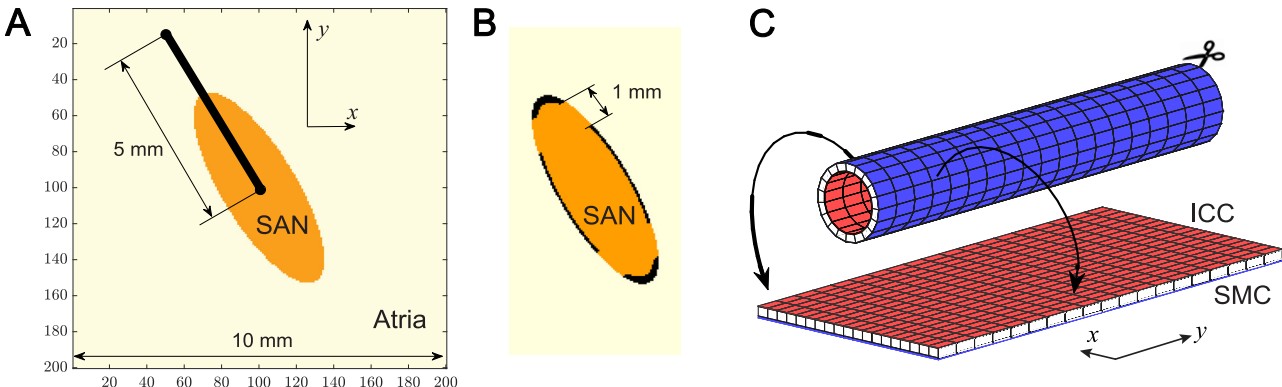

**Fig 3. Schematics of the simulation examples.** A. 2D SAN model with the pAP-AP cells. B. SAN structure with a border of passive tissue and exit pathways. C. 3D intestine model with the pCN-CN cells.

## 2D SAN model

The primary natural cardiac pacemaker, SAN, consists of a small area of specialized cells situated in the right atrium (the right upper chamber of the heart). The SAN dysfunctions may result in dangerous cardiac arrhythmias. The mechanisms and processes involved in the latter are very complicated and may be difficult or nearly impossible to explain without the help of computer modeling.

As application examples for the pAP-AP coupled system, we performed simulations of simplified 2D SAN models illustrating the effect of SAN—atrium coupling on the pacemaking behavior. The SAN model consists of a rectangular area of 10 mm by 10 mm ($200 \times 200$ mesh, $\Delta x = 0.05$ mm spatial step size) of atrial tissue represented by AP model cells (Fig 3A). The pacemaker was defined as an elliptically shaped area of pAP cells in the center of the rectangular with half-axes 3 and 1 mm, and long SAN axes 30° off the fiber direction. These dimensions approximately correspond to the canine heart [57].

Two different SAN structure types were considered. Type 1, without insulating border and exit pathways (Fig 3A), was similar to that demonstrated in work [56]. The whole tissue was anisotropic with the ratio of diffusion coefficients $D_y$: $D_x = 1$:0.208 (1.2: 0.25 conductivity ratio in [56]) along the fiber longitudinal $y$ and transverse $x$ directions, respectively. Two different values of diffusion coefficient $D_y = 0.090$ and 0.048 mm$^2$ms$^{-1}$ were used. The parameters used in the simulations are given in Table 2.

The whole model geometry in type 2 was similar to the first but with isotropic tissue ($D_y$: $D_x = 1$:1). The pacemaker area was surrounded by borders of passive tissue with four symmetric exit pathways of 1 mm width (Fig 3B), following previous studies [58–60]. The diffusion coefficient $D_A = 0.160$ mm$^2$ms$^{-1}$ was fixed in the atrium, while two different values $D_S = 0.060$ and 0.052 mm$^2$ms$^{-1}$ were set for the SAN pacemaker region (Fig 3B). The passive tissue was defined as [28]:

$$\frac{\partial u}{\partial t} = -c_t S u + \nabla \cdot (\mathbf{D}\nabla u) , \qquad (10)$$

where $S = 26$ is the tissue conductivity. The tissue parameters within the exit pathways were the same as those of the SAN pacemaker.

In this simple SAN structure, we incorporated neither diffusion gradients nor electrical heterogeneity of pacemaker cells, in contrast to the widely adopted approach in the simulations with ion-channel models [53, 59].

### 3D intestine model

There are two main layers of different cell types in the intestine. The first layer of specialized pacemaker cells, termed interstitial cells of Cajal (ICC), produces slow propagating electrical waves. The ICC cells synchronize to the highest frequency within the layer. This electric activity controls the contractile stress exerted by the second layer of smooth muscle cells (SMC) of the intestinal tissue. Both layers of ICC and SMC are electrically connected via electrotonic coupling.

In the second application example for the proposed pacemaker models, we considered a simple 3D electrophysiological model of the small intestine. As a reference, we considered results from the papers [23, 24, 61], in which the FHN and ion-channel models were used. In contrast to the works, we described ICC and SMC layers with the pCN and CN model cells, respectively, to demonstrate the broad applicability of the pCN model. It has been demonstrated that the excitable MS model is apt to spontaneous excitations at some conditions, with its modification proved to be robust to such pacemaker behavior [19]. Using the pCN model with convenient frequency control in combination with the robust CN model instead of the MS and FHN models may improve the behavior of the pertinent computational tissue models.

Similar to [23, 62], our intestine model geometry is presented by a long two-layer tube, cut along its axle $y$ on one side and stretched to a dual-layer plane (Fig 3C). The blue and red surfaces in Fig 3C represent external SMC and internal ICC layers, respectively. The simulation domain for both layers has dimensions $N_x \times N_y = 176 \times 4800$, with uniform spatial mesh size $\Delta x = 0.25$ mm, which corresponds to the 1200 mm long tube (one half of that in [23]) with the mean circumference of 44 mm. Such a simple tube geometry corresponds to the anatomy of the small intestine in general and to previously reported values of the intestine of average-size animals like dogs or rabbits [62–64].

The following set of ODEs describes electrical dynamics in the intestine layers for transmembrane potentials $u^I$ of ICC and $u^M$ of SMC:

$$\frac{\partial u^I}{\partial t} = I_{tot}^I + d^{IM}(u^M - u^I), \tag{11}$$

$$\frac{\partial u^M}{\partial t} = I_{tot}^M + d^{IM}(u^I - u^M), \tag{12}$$

where $I_{tot}^I$ and $I_{tot}^M$ represent the right-hand side of the Eq 6 for ICC and SMC layers, respectively, and last terms in the right-hand side of Eqs 11 and 12 describe the electrotonic coupling between the layers with coupling coefficient $d^{IM} = 6 \times 10^{-3}$ ms$^{-1}$. Conduction within the layers was considered isotropic with the diffusion coefficients $D^I = 5 \times 10^{-5}$ mm$^2$ms$^{-1}$ and $D^M = 8 \times 10^{-4}$ mm$^2$ms$^{-1}$ for the ICC and SMC, respectively. Each of Eqs 11 and 12 is accompanied by the three corresponding equations for slow variables $h^I$ and $h^M$ similar to Eqs 7–9.

Individual isolated ICC oscillate at different intrinsic frequencies, with spatial frequency gradient in the longitudinal direction from the pylorus (first part of the duodenum, left side in Fig 3B) toward the ileum (right side in Fig 3B). To create the frequency gradient in the ICC layer, for Eq 11 we set up an exponential distribution of the parameter $b_{CN}$ along the $y$ axis (and constant along the $x$ axis) [23]:

$$b_{CN}(i) = 0.4 + 1.3 \cdot \exp\left(-\frac{i\Delta x}{680}\right), \tag{13}$$

where $i$ is the cell index counting from the duodenum. Eq 13 yields intrinsic oscillation frequency distribution of ICC-SMC coupled pairs from 17.5 to 8 cpm along full-length of 2400 mm small intestine, or 17.5—10.5 cpm for the upper half [64] (see Fig 9B).

To demonstrate the appearance of intestinal dysrhythmias in a similar way as in [23, 62], a temporal conduction block ($u^I = u^M = 0.001$, $h^I = h^M = 0.5$, one time step long) was induced at $t = 5100$ s to the rectangular area at both ICC and SMC layers with the width $l_y = 40$ mm, height $l_x = 22$ mm, and origin at $x_0 = 0$ and $y_0 = 580$ mm. Neither the electromechanical [39] nor thermodynamical [23] coupling was included in the model for the sake of simplicity.

## Numerical methods

All simulations were performed with MATLAB (R2021b) on a usual desktop computer with AMD Ryzen 9 3950X CPU. For the acceleration of 3D intestine simulations, NVidia RTX 3090 GPU was used. We employed both the explicit forward Euler (FE) method for the preliminary simulations and the implicit backward Euler (BE) method for final results to solve the ODE systems. In the BE method, the absolute tolerance was set to $1 \times 10^{-7}$ with the maximum number of iterations in the inner loop 20. The latter was not exceeded in all simulations.

No-flux Neumann boundary conditions were applied in 1D, 2D, and 3D simulations except the periodic boundary conditions along the $y$ axes in the 3D intestine model. Equilibrium points for bifurcation diagrams were calculated with MatCont software [65]. The parameters for the AP and pAP, CN and pCN models used in the simulations are listed in Tables 1–3, respectively. The initial conditions for the models were chosen to be $u(0) = 0.01$, $v(0) = 0.01$ for the AP and pAP, and $u(0) = 0.01$, $h(0) = 0.5$ for the CN and pCN models.

To estimate the simulations' accuracy, we compared the results for single-cell simulations with the FE method at different time steps $\Delta t$ with that calculated with the unconditionally stable BE method with a small time step $\Delta t = 0.0001$ ms. The relative norms

$$L_2 = \frac{\|u^{BE} - u^{FE}\|_2}{\|u^{BE}\|_2} \text{ and } L_\infty = \frac{\|u^{BE} - u^{FE}\|_\infty}{\|u^{BE}\|_\infty} \tag{14}$$

(for a single cycle) and relative frequency error (for 60 s runs), as well as a speedup of calculations, are given in Table 4. For 1D coupled pacemaker-excitable systems, the maximum relative frequency error with $\Delta t = 0.1$ ms was also about 0.16%. As seen from Table 4, in most cases of the simulations with the pAP and pCN models, the FE and BE methods with a time step of 0.1–0.01 ms would be enough to obtain reasonable accuracy (see also S1 Fig). The

**Table 1. Parameters for the pAP and AP models used in the 0D (isolated cell) and 1D simulations.**

| Cell | $k$ | $a$ | $\varepsilon_0$ | $\mu_1$ | $\mu_2$ | $b_{AP}$ | $\Delta t$ (ms) |
|------|-----|-----|------------------|---------|---------|----------|------------------|
| pAP (0D) | 8 | 0.13 | 0.002 | 0.2 | 0.3 | 0—0.08 | 0.01 |
| pAP (1D) | | | | | | 0—0.5 | 0.1—0.01 |
| AP (1D) | | | | | | -0.13 | |

**Table 2. Parameters for the pAP and AP models used in the 2D SAN simulations.**

| Cell | $k$ | $a$ | $\varepsilon_0$ | $\mu_1$ | $\mu_2$ | $b_{AP}$ | $\Delta t$ (ms) | $\Delta x$ (mm) |
|------|-----|-----|------------------|---------|---------|----------|------------------|------------------|
| pAP (SAN, type 1) | 8 | 0.13 | 0.002 | 0.2 | 0.5 | 0.120 | 0.002 | 0.05 |
| pAP (SAN, type 2) | 12 | | | | | 0.133 | | |
| AP (Atrium) | 8 | | 0.045 | | 0.3 | -0.13 | | |

**Table 3. Parameters for the pCN and CN models used in the 0D (isolated cell), 1D, and 3D simulations.**

| Cell | $\tau_{in}$ (ms) | $\tau_{out}$ (ms) | $\tau_{open}$(ms) | $\tau_{close}$ (ms) | $u_s$ | $u_{gate}$ | $b_{CN}$ | $\Delta t$ (ms) | $\Delta x$ (ms) |
|---|---|---|---|---|---|---|---|---|---|
| pCN (0D) | 0.3 | 6.0 | 120 | 150 | 0.05—0.55 | -0.1—0.35 | 0.05—0.6 | 0.01 | - |
| pCN (1D) | | | | | 0.15 | -0.05 | 0.1—12 | 0.1—0.01 | - |
| CN (1D) | | | | | 0.01 | 0.13 | -0.13 | | |
| pCN (ICC) | 16 | 200 | 1500 | 1800 | 0.20 | -0.05 | 0.62—1.70 | 2.5 | 0.25 |
| CN (SMC) | 11 | | | | 0.01 | 0.10 | -0.10 | | |

**Table 4. Comparison of the accuracy of the results obtained with FE and BE methods at different time steps $\Delta t$ (Eq 14).**

| Cell | FE, $\Delta t$ = 0.1 ms | | | | FE, $\Delta t$ = 0.01 ms | | | | FE, $\Delta t$ = 0.001 ms | | | |
|---|---|---|---|---|---|---|---|---|---|---|---|---|
| | $L_2$ | $L_\infty$ | Freq. | Speedup | $L_2$ | $L_\infty$ | Freq. | Speedup | $L_2$ | $L_\infty$ | Freq. | Speedup |
| pAP | 0.25% | 0.93% | 0.17% | 707 | 0.14% | 0.38% | 0.017% | 218 | 0.01% | 0.01% | 0.002% | 36 |
| pCN | 0.53% | 2.03% | 0.16% | 1040 | 0.13% | 0.45% | 0.016% | 123 | 0.08% | 0.28% | 0.002% | 14 |

performance of both FE and BE methods implemented in the CHASTE open source software package was also demonstrated in work [66].

At the same time, the standard stability criterion

$$\frac{D\Delta t}{\Delta x^2} = d\Delta t < \frac{1}{2N},\qquad(15)$$

where $N$ is the dimension of the simulation domain, should be taken into account as well [4]. This criteria was fulfilled for both SAN (at the highest $D$ = 0.160 mm$^2$ms$^{-1}$) and intestine ($D^I$ = $8 \times 10^{-5}$ mm$^2$ms$^{-1}$) simulations. In the 2D and 3D simulations, we used the same time and space discretizations for both FE and BE methods (Tables 2 and 3), and the obtained results were visually similar for both methods.

## Results and discussion

### Single-cell pacemaker dynamics

The phase portraits and nullclines of excitable and pacemaking model variants are presented in Fig 2. Three action potential waveforms are demonstrated in Fig 2A.3 and 2B.3. They correspond to the limit cycles with EPs shown by squires of the same color in Fig 2A.2 and 2B.2, calculated with different values of the parameters $b_{AP}$ and $b_{CN}$, respectively.

The absence of undershoot of the action potential amplitude in the pAP and pCN models (in contrast to the FHN model [9, 10]) makes them specifically suitable for utilization as secondary pacemakers and in the cardiac simulation systems with a limited number of elements, similar to proposed in the works [44–46].

Figs 4 and 5 demonstrate the dependence of single-cell dynamic characteristics of the pAP and pCN models on various bifurcation parameters. The trajectories of the EPs are shown by dashed lines in the left columns of Figs 4 and 5.

All the bifurcation parameters affect the intrinsic oscillation frequency (central columns of Figs 4 and 5). For pAP, the highest variation in the frequency was observed with $b_{AP}$ (with almost linear dependency, Fig 4A.2) and $I_{ext}$ (Fig 4E.2). The former allows the most convenient control of the frequency, while the latter indicates the strong sensitivity of the model to the external coupling strength. The variation of POP and MDP is strongest also for $b_{AP}$ and $I_{ext}$.

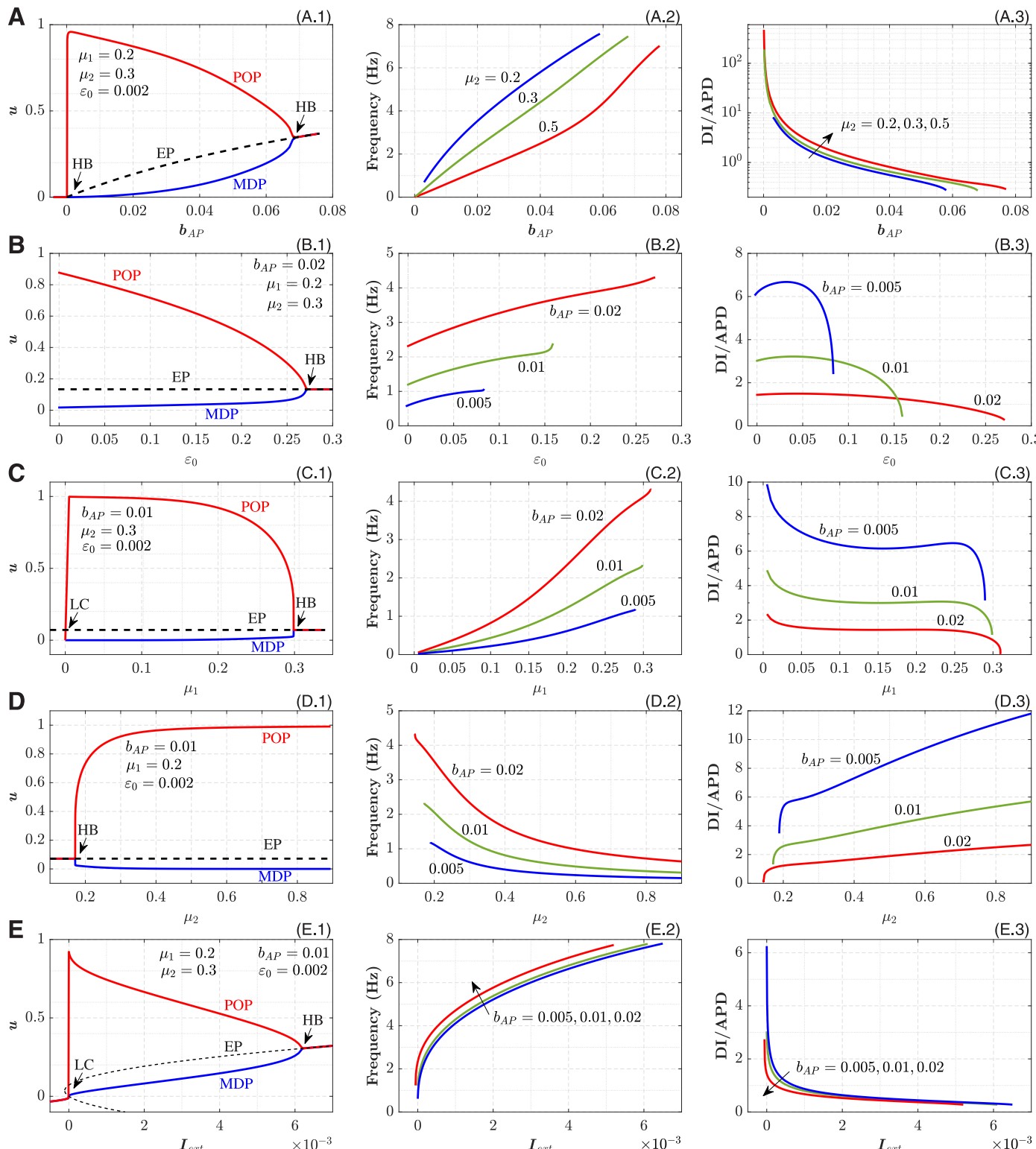

**Fig 4. Dependence of pAP cell characteristics on various parameters.** A. On the bifurcation parameter $b_{AP}$. B. On the parameter $\epsilon_0$. C. On the parameter $\mu_1$. D. On the parameter $\mu_2$. E. On the external current $I_{ext}$. Left panels correspond to the bifurcation diagrams, central—to the calculated frequencies, and right panels—to the calculated DI on APD ratios.

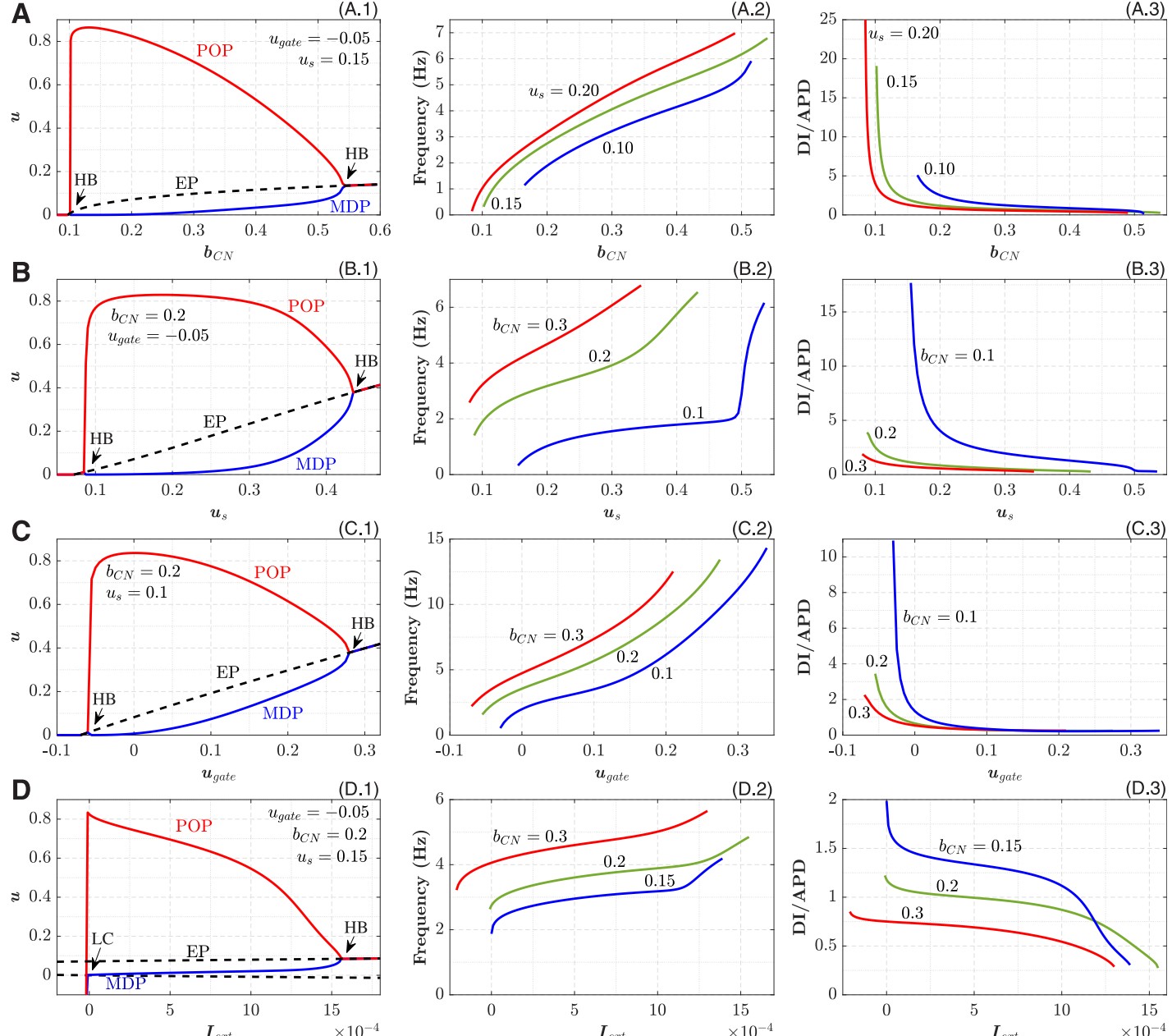

**Fig 5. Dependence of pCN cell characteristics on various parameters.** A. On the bifurcation parameter $b_{CN}$. B. On the slope factor $u_s$, C. On the parameter $u_{gate}$. D. On the external current $I_{ext}$. Left panels correspond to the bifurcation diagrams, central—to the calculated frequencies, and right panels—to the calculated DI on APD ratios.

The increase of the parameter $\mu_2$ in the 0.1–0.5 range decreases the frequency, and higher values of $\mu_2$ seem to be impractical (Fig 4D.2). For the pCN model, significant variation of POP and MDP was observed for all bifurcation parameters (left columns in Fig 5).

The dependencies of DI/APD ratios on the bifurcation parameters are shown in the right columns of Figs 4 and 5. The ratios supplement the intrinsic frequency characteristics demonstrating DI and APD contributions into the cycle length. With decreasing DI/APD ratio, the steepness of the restitution curve (APD on DI) increases. At some combination of the

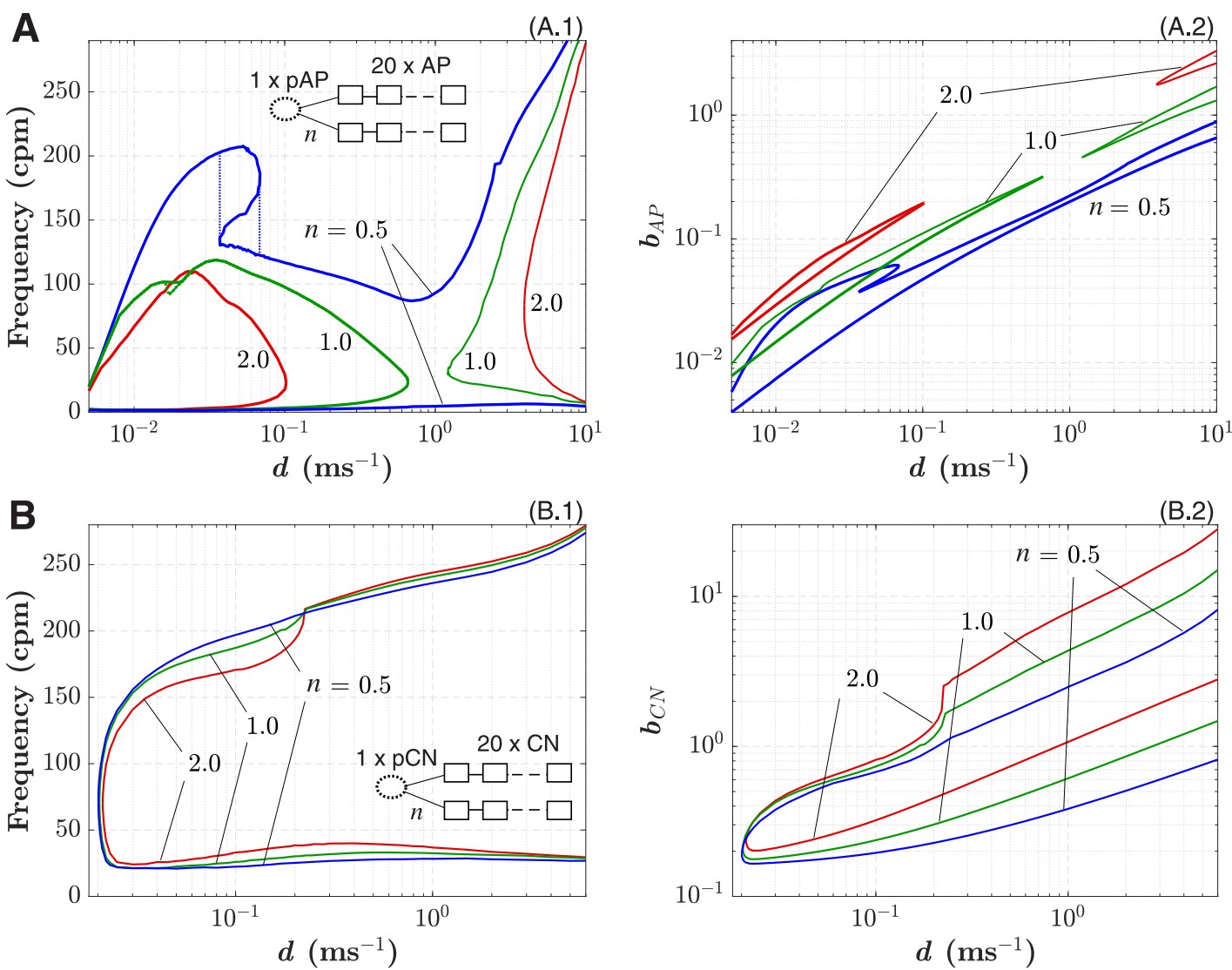

**Fig 6. A. 1D pAP-AP coupled system.** A.1. Minimal and maximal 1:1 synchronized frequency dependence on the coupling coefficient $d$ for the pAP cell coupled with $n = 2$ (red), 1 (green), and 0.5 (blues) strands of 20 AP cells. Inset illustrates a schematic representation of the 1D pAP-AP coupled system. Ellipse symbolizes pacemaker cell and rectangles—excitable cells. A.2. Minimal and maximal values of the parameter $b_{AP}$ vs $d$ corresponding to panel A.1. **B. 1D pCN-CN coupled system**. B.1. Minimal and maximal 1:1 synchronized frequency dependence on the coupling coefficient $d$ for the pCN cell coupled with $n = 2$ (red), 1 (green), and 0.5 (blue) strands of 20 CN cells. Inset illustrates a schematic representation of the 1D pCN-CN coupled system. B.2. Minimal and maximal values of the parameter $b_{CN}$ vs $d$ corresponding to panel B.1.

parameters the ratios are always higher than unity (see, for example, $b_{AP} = 0.005$ curve in Fig 4C.3). For the pCN model, this takes place only for $I_{ext}$ dependence (Fig 5D.3).

In the pCN model, $b_{CN}$ (Fig 5A.2) and $u_{gate}$ (Fig 5C.2) are the parameters with the highest variation of intrinsic frequency. The parameter $b_{CN}$, similar to $b_{AP}$, allows obtaining very low oscillation rates (less than 0.1 Hz). The influence of $I_{ext}$ (Fig 5D.2) on the frequency is relatively weak, thus the pCN model looks less sensitive to the coupling strength (compare with Fig 4E.2, see also Fig 6).

Both models demonstrated notably wide ranges of intrinsic frequencies: 0.007—7.6 Hz for pAP, 0.14—14 Hz for pCN, corresponding to about 0.4—450 and 8.4—840 counts per minute (cpm), respectively. Such a broad frequency span allows the implementation of the models for

the simulation of various organs of different animal species at normal and pathological conditions.

For the pCN model, the frequency and, in particular, DI/APD ratios depend also on the time constants $\tau_{in}$, $\tau_{out}$, $\tau_{open}$, and $\tau_{close}$. The influence of the constants on pCN characteristics is similar to that for the original and modified MS models [18, 19, 50].

## Dynamics of 1D coupled pacemaker-excitable system

Depending on the coupling (diffusion coefficient), the pacemaker-excitable system may be either in a fully synchronized (1:1) regime, when all excitable cells in a strand follow the driving frequency or in an asynchronous/chaotic regime when not all of the pacemaker action potentials are able to propagate to the end of the strand [56, 67].

Fig 6A.1 and 6B.1 demonstrate dependencies of minimal and maximal frequencies of complete 1:1 synchronization and corresponding minimal and maximal values of parameters $b_{AP}$ and $b_{CN}$ on the coupling coefficient $d = D/\Delta x^2$ for the single pAP and pCN model cells coupled with $n$ = 0.5, 1, 2 strands of 20 AP and CN excitable cells, respectively.

For the pAP-AP coupled system, the complete synchronization was confined in two separate areas created by interlocks of minimal and maximal frequency curves (Fig 6A.1). The areas partially merged in the case of $n$ = 0.5. Similar corresponding separate areas of the parameter $b_{AP}$ are seen in Fig 6A.2. This behavior indicates that at certain values of $d$, complete 1:1 synchronization in the pAP-AP system with a heavy load can not be obtained, at least with the model parameters used.

The existence of the two separate areas in the coupling dependence characteristics is associated with two possible variants of intersections between branches of the parabolic nullclines of the pAP model (Eqs 2 and 3). The areas of complete synchronization became smaller with an increasing number of strands $n$.

We also observed a pronounced hysteresis in the case with $n$ = 0.5 (and a very small one with $n$ = 1.0) of the maximal synchronized frequency characteristic at low coupling strength. Such effect seems similar to that demonstrated in the simulations of electrically coupled pacemaker and non-pacemaker cells with the VCD model [68].

In the pCN-CN coupled system, changes in the complete synchronization areas for different $n$ were insignificant, and maximal synchronized frequency monotonically increased (Fig 6B.1). Fig 6B.1 and 6B.2 demonstrate much wider synchronization areas for both frequency and parameter $b_{CN}$. This may be attributed to the higher energy capacity of the pCN over pAP cell with conventional model parameters. For both models increasing the number of strands and increasing coupling strength required rising values of the corresponding parameter $b$ to maintain complete synchronization (Fig 6A.2 and 6B.2).

## Synchronization behavior with fixed intrinsic pacemaker frequency

Fig 7 demonstrates the synchronization behavior of the pAP-AP and pCN-CN 1D systems with increasing coupling at fixed bifurcation parameters (fixed intrinsic pacemaker frequency). In Fig 7A and 7B, one can observe the existence of a certain threshold value of the coupling coefficient $d$, below which the pacemaker frequency was much higher than that of the coupled strand(s) of excitable cells. The threshold values decreased with increasing coupling load, i.e., with the increasing number of strands $n$. With the reduction of $d$ from the threshold value, the pacemaker-excitable frequency ratios increased significantly with abrupt jumps, as observed in most simulated cases.

Above the threshold values of $d$, complete synchronization occurred with a smooth, gradual drop of the frequency in the pAP-AP system (Fig 7A), while in the pCN-CN system, it reaches

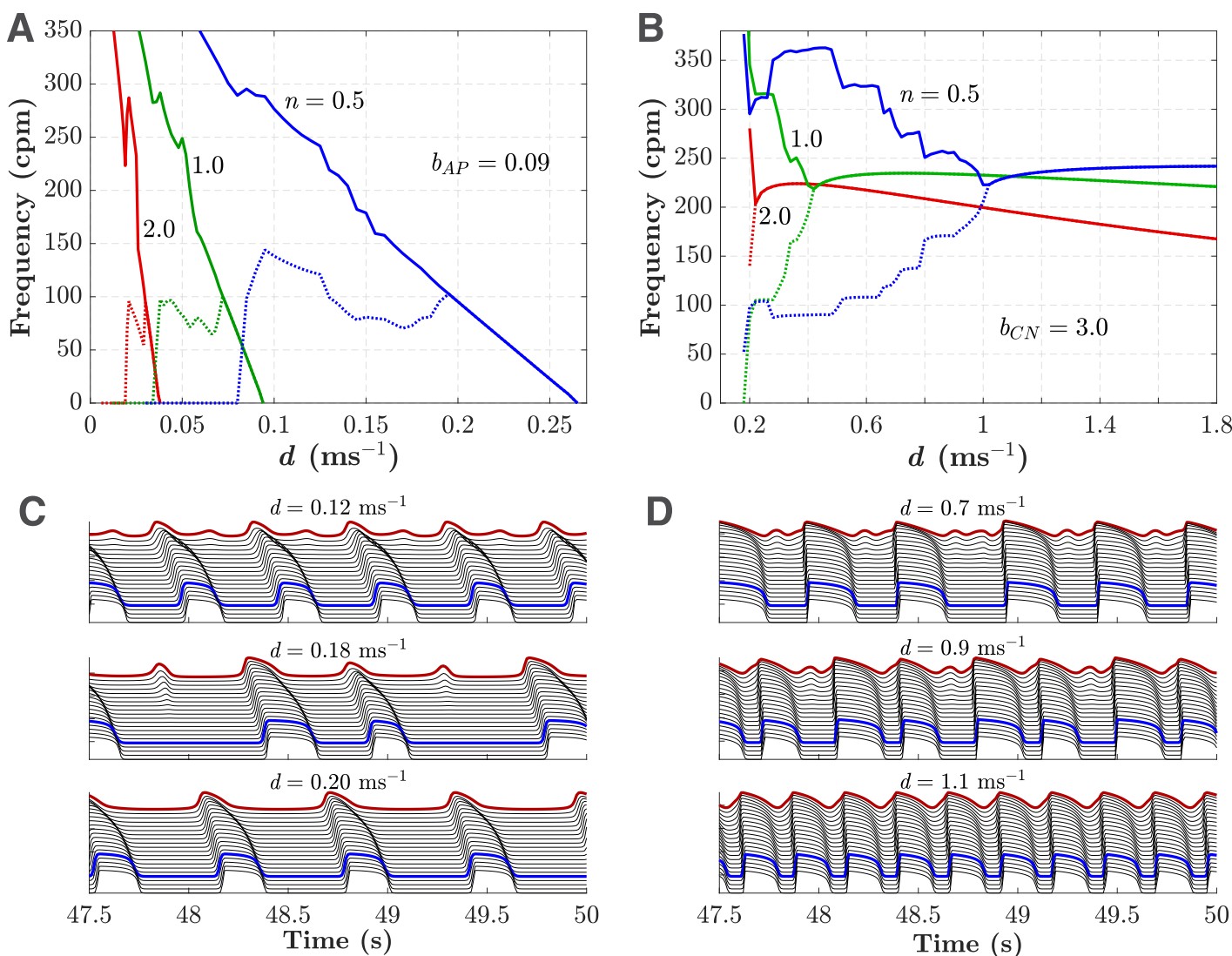

**Fig 7. Synchronization in the simplified 1D coupled model systems with fixed parameters *b*.** Dependence of synchronization in the pAP-AP (A) and pCN-CN (B) systems with a different number of excitable cell strands *n* on the coupling parameter *d*. Solid lines correspond to the pacemaker frequency and the dotted lines—to the frequency at the 16th excitable cell in the strand(s). C. Action potentials for the pAP-AP system for the number of strands *n* = 0.5. The bold red and blue lines mark the action potentials of the pacemaker cell and 16th cell of a strand, respectively. D. The same as in panel C, but for the pCN-CN system.

a maximum before final fall-off (Fig 7B). Fig 7C and 7D show action potentials of the cells in both coupled systems with the number of strands *n* = 0.5 for three different values of the coupling coefficient *d*. One can see the transition from incomplete to complete synchronization with increasing *d*. Also, for higher loads, the complete synchronization took place at lower *d*. The pCN-CN coupled system reached the complete synchronization state at much higher values of *d* than the pAP-AP and higher frequencies.

## 2D simulation of SAN

Fig 8 shows the calculated spatio-temporal distributions of the transmembrane potential *u* from the central SAN region to peripheral atrial tissue (along the bold black line in Fig 3A) and activation sequences for different diffusion coefficients *D* (for the SAN model variant

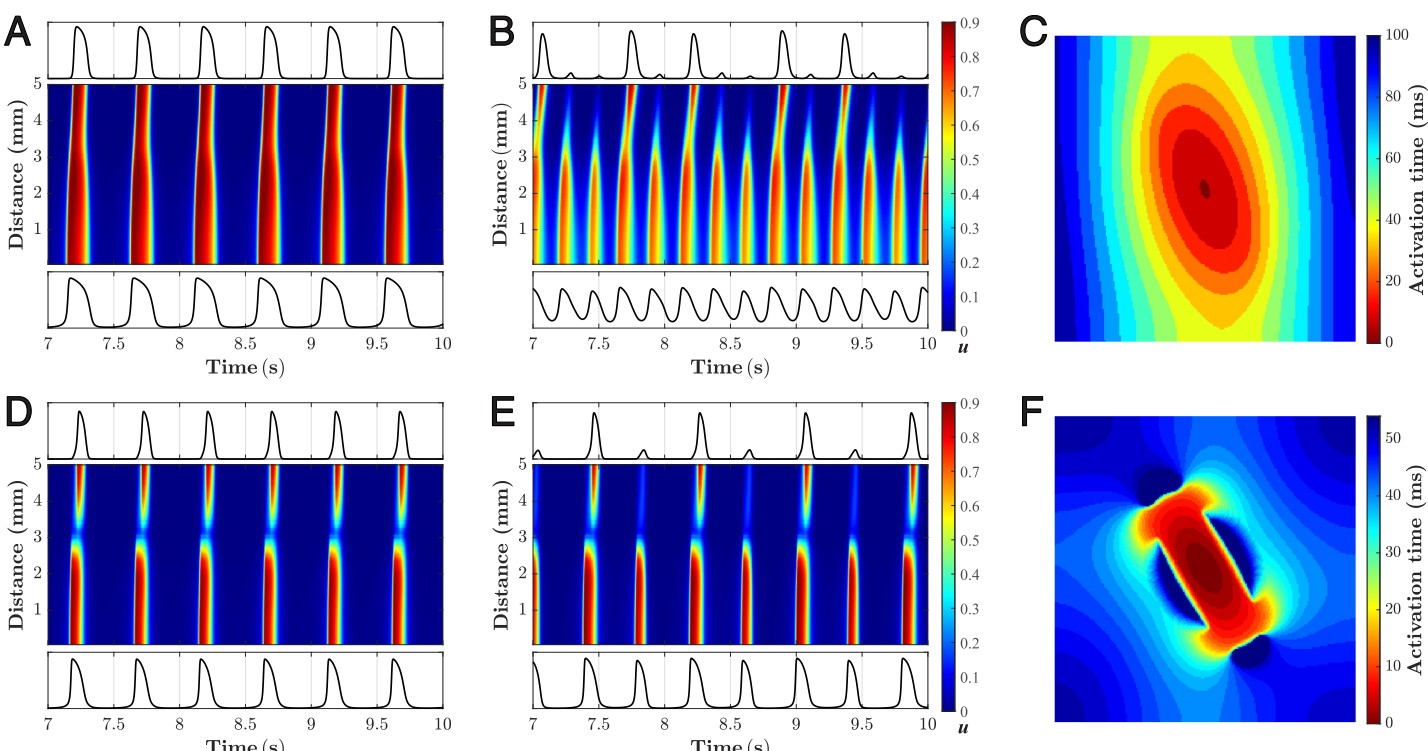

**Fig 8. 2D simulation of SAN.** Calculated time sequences of the transmembrane potential $u$ along the bold black line in Fig 3A and activation sequences for SAN structure without (A–C) and with (D–F) walls of passive tissue. A. For diffusion coefficient $D = 0.090$ mm$^2$ms$^{-1}$. B. $D = 0.048$ mm$^2$ms$^{-1}$. C. Activation sequence corresponding to the case shown in panel A. D. $D_A = 0.160$ mm$^2$ms$^{-1}$, $D_S = 0.060$ mm$^2$ms$^{-1}$. E. $D_A = 0.160$ mm$^2$ms$^{-1}$, $D_S = 0.052$ mm$^2$ms$^{-1}$. F. Activation sequence corresponding to the case shown in panel D.

without insulating border, Fig 8A–8C) and for different diffusion coefficients $D_S$ in the SAN pacemaker area (for the SAN model variant with insulating border and exit pathways, Fig 8D–8F).

With relatively strong coupling ($D = 0.090$ mm$^2$ms$^{-1}$) SAN and atrial excitations were synchronized with a slow pacing rate (about 120 cycles per minute, Fig 8A, see also S1 Video). When the coupling became too weak ($D = 0.048$ mm$^2$ms$^{-1}$), the atrial tissue failed to respond on every SAN excitation with 2:5 ratio (Fig 8B, S2 Video). The simulation results of this example are close to that demonstrated previously with the similar 2D SAN structure using complex ion-channel cell models with sufficient and weak SAN-atrial coupling [56]. Fig 8C (and S3 Video) shows activation sequence corresponding to $D = 0.090$ mm$^2$ms$^{-1}$.

In the other SAN structure variant, the high value of diffusion coefficient $D_S = 0.060$ mm$^2$ms$^{-1}$ resulted in complete SAN-atria synchronization (Fig 8D, and S4 Video). When $D_S$ decreased to $0.052$ mm$^2$ms$^{-1}$, every second action potential propagated from the SAN center to the border failed to depolarize the atria (Fig 8E, and S5 Video). Due to the block zones placed on the SAN ellipse vertexes, the gap appeared in the transmembrane potential propagation path along the bold line in Fig 3A. Fig 8F (and S6 Video) shows activation sequence corresponding to $D_S = 0.060$ mm$^2$ms$^{-1}$. The effect of the reduced diffusion coefficient on the SAN-atria synchronization looks similar for both variants despite differences in the SAN structures. As seen from Fig 8, in the simulation of the SAN-atria system with pAP-AP coupled cell models, a stable pacing function can be obtained regardless of the presence or absence of block zones of passive tissue around the SAN pacemaker.

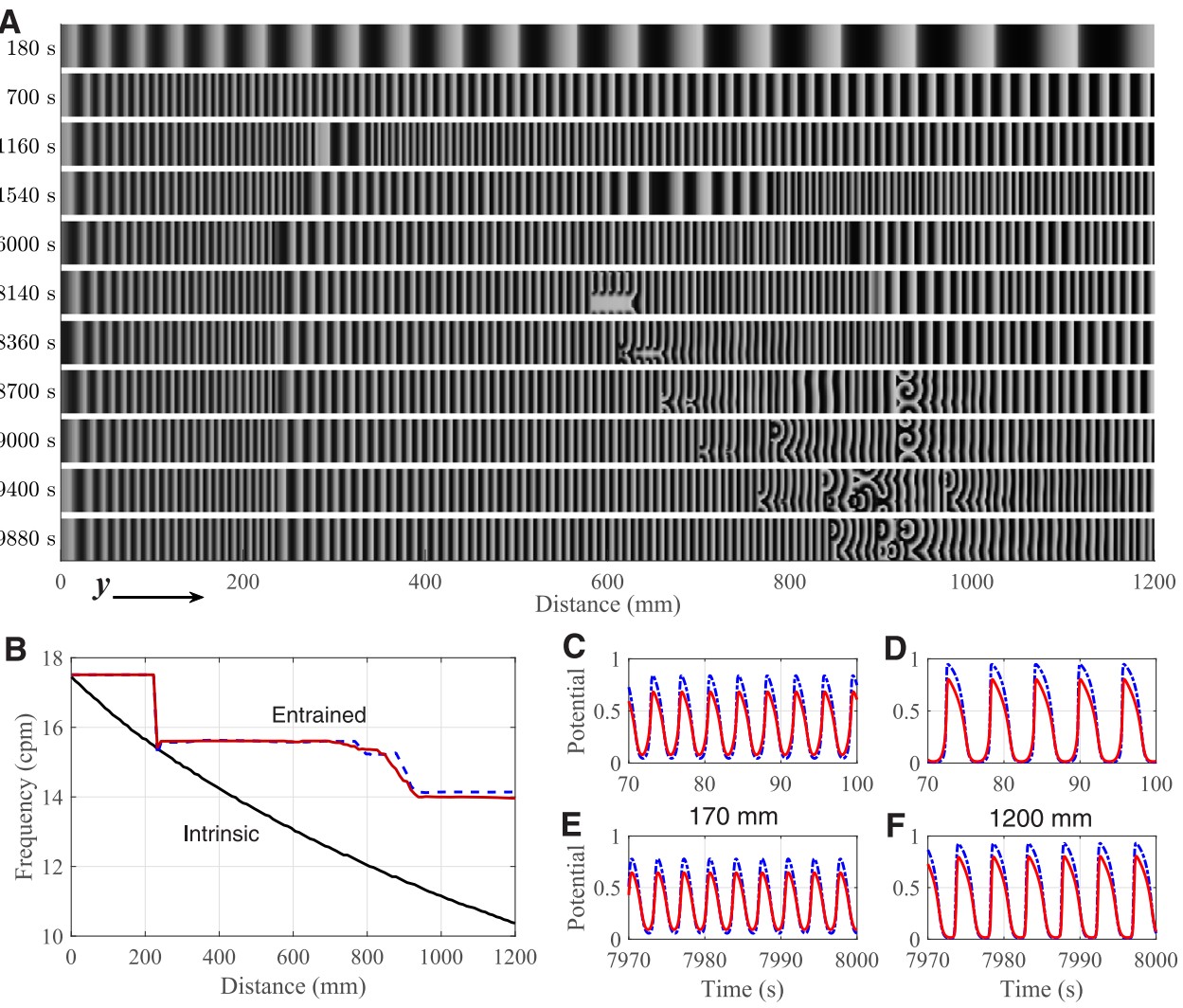

**Fig 9. 3D simulation of intestine.** A. Snapshots of spatial distributions of the transmembrane potential $u^I$ in the ICC layer at different time moments $t$. Brighter areas correspond to higher potential. From top to bottom: (1–5) formation of constant frequency plateaus, (6–11) onset and evolution of intestinal dysrhythmia pattern due to temporary conduction block induced at $t = 8100$ s to both ICC and SMC layers. B. Distributions of the intrinsic frequency ($D^I = D^M = 0$, solid black line), entrained frequency of oscillators at $t = 8000$ s (solid red line), and at the end of the simulation ($t = 11600$ s, dashed blue line). C–D. $u^I$ (red solid lines) and $u^M$ (blue dot-dashed lines) action potentials at the distance of 170 mm (C) and 1200 mm (D) at the beginning of the simulation ($t \simeq 100$ s). E–F. The same as in panels C and D, but at $t \simeq 8000$ s.

## 3D simulation of intestine

The results of the 3D intestine simulations are presented in Fig 9 (see also the animated version in S7 and S8 Videos). Fig 9A shows spatial distributions of the transmembrane potential $u^I$ in the ICC layer at different time moments.

Slow waves of ICC were generated in the leading pacemaker region and traveled distally, i.e., from left to right along the small intestine (Fig 9A). The initial distribution of the intrinsic frequency of each ICC-SMC coupled pair ($D^I = D^M = 0$) is shown in Fig 9B. The lengths of the waves gradually decreased in time ($t = 180$ and 700 s), reaching complete entrainment at about $t = 8000$ s (S7 Video). During the transient process ($t = 1160$, 1540, and 6000 s), plateaus with constant frequency were formed (Fig 9B). The plateau boundaries correspond to the phase dislocations, at which some waves were occasionally dropped ($y \simeq 240$ mm and 940 mm, Fig 9A).

The formed plateaus of entrained frequencies are in agreement with the recent experimental findings [69].

The sixth panel from the top ($t$ = 8140 s) in Fig 9A demonstrates the situation right upon the induction of the temporary conduction block ($t$ = 8100 s). The block overlapped a few consequent waves, initiating intestinal dysrhythmia in the form of turbulent perturbations ($t$ = 8180, 8360, and 8700 s) [23, 62]. The perturbation area, which appeared in the middle of the second frequency plateau, traveled distally toward the ileum until reaching the point of the phase dislocation, where it persisted the rest of the simulation ($t$ = 9400 s and 9880 s) in the form of spiral wave or rotor activity [62]. The slow action potential waves in the SMC layer followed the waves in the ICC layer with higher amplitude (Fig 9C–9F). The ICC $u^{\mathrm{I}}$ (red solid lines) and SMC $u^{\mathrm{M}}$ (blue dot-dashed lines) action potentials at the distance of 170 mm and 1200 mm from the duodenum at the initial moment ($t \simeq 100$ s) are demonstrated in Fig 9C and 9D, and at $t \simeq 8000$ s in Fig 9E and 9F, respectively.

The obtained results presented in Fig 9 demonstrate the capability of the phenomenological model based on the pCN-CN system to reproduce physiological processes in the intestine [61, 69].

## Limitations and general remarks

Since this paper aimed to demonstrate the applicability of the pAP and pCN phenomenological models for simulations of various physiological systems, we did not precisely adjust the parameters of the SAN and intestine model cells to the existing experimental data. The results in Figs 8 and 9 are presented for illustration purposes and not aimed at a detailed study of these organs. In particular, in the SAN type 2 model (Fig 3B), the tissue of the borders surrounding the pacemaking area is of passive type, blocking the spread of excitation [28]. Such tissue, being not exactly of an electrically insulating type [59, 60], causes electrotonic interactions with the surrounding SAN and atrial tissues. Better selection of the boundary tissue properties can be beneficial for more realistic simulations of the SAN and its exit pathway functions. Moreover, additional tuning of the parameters to clinical data might be necessary for accurate simulations, in particular, to construct patient-specific models. This can be realized by applying, for example, a robust and clinically tractable protocol and fitting algorithm [19] for characterizing cardiac electrophysiology properties by simplest two-variable cell models, such as the pAP-AP and pCN-CN coupled systems.

For the simulations where the differences of resting and/or peak levels are necessary, the MDP and POP values of the considered models can be modified. The addition of a constant to the term $u$ and the replacement of unity in the term $(1 - u)$ allow shifting of the MDP and POP levels, respectively, though the exact resulting values of the latters cannot be determined directly (see S2 Fig for the details). Also, such modification may require an adjustment of the intrinsic frequencies (for example, with the parameters $b_{AP}$ and $b_{CN}$). The above modifications of the MDP and POP together with modulation of the intrinsic frequency with the parameters $b$ may allow simulation of a kind of tonic bursting [29]—a firing behavior in which a neuron cell fires a certain number of spikes on the top of the plateau and is silent for a certain amount of time. Though, compared to the variety of specific neuronal cell models [29], the pAP and pCN models may not be the best choice for modeling of neuronal systems.

Another well-known disadvantage of the phenomenological models like pAP-AP and pCN-CN is their limited ability to represent action potential morphology under varying physiological conditions, e.g., the effect of variation of concentration of particular ions, which is necessary for simulations of complex cardiac diseases such as ion channelopathies. However, the utilization of the considered pacemaker models can be beneficial for the development of

simplified real-time simulation systems (e.g., personalized medicine [30, 31] and inverse cardiac modeling [32]), deep learning [43], as well as for medical device testing platforms [44, 46]. The simplified models can also be used in preliminary simulation setups for hypothesis testing and verification, as well as model design and tuning before implementing complex and computationally expensive ion-channel models [34].

## Conclusion

In this work, we considered pacemaking variants of the Aliev-Panfilov and Corrado two-variable excitable cell models. We studied the main features of single-cell nonlinear dynamics of the models, their synchronization behavior in one-dimensional coupled pacemaker-excitable setups, including regions of complete synchronization, and the relationship between pacemaker frequency and overall coupling. Also, we performed simulations of the simplified 2D sinoatrial node and 3D intestine models employing the considered pacemakers. The obtained spatio-temporal dynamics of the transmembrane potentials in the 2D and 3D models are in general agreement with that demonstrated previously for identical schematics with different cell models.

An essential feature of the pacemaker models is that they do not include any additional equations for currents, thus having the same number of variables as the original excitable models, allowing a simple uniform description of the whole pacemaker-excitable system. We believe that the pacemaker variants of the Aliev-Panfilov and Corrado models can be used for computationally efficient electrophysiological modeling of tissues that include primary and subsidiary pacemaking cells, allowing the development of models for whole organs, various species, patient-specific medicine, and real-time testing and validation of medical devices.

## Supporting information

**S1 Fig. Action potentials and phase portraits of the pAP and pCN models.** Comparison of the accuracy of Forward Euler and Backward Euler methods. The MATLAB and CellML codes are available at https://github.com/mryzhii/Simplified-pacemaker-cell-models.
(PDF)

**S2 Fig. Shifting of the POP and MDP levels in the pAP and pCN models.**
(PDF)

**S1 Video. Animation of the transmembrane potential $u$ in the SAN model structure (type 1).** Corresponds to Fig 8A for diffusion coefficient $D = 0.090$ mm$^2$ms$^{-1}$.
(MP4)

**S2 Video. Animation of the transmembrane potential $u$ in the SAN model structure (type 1).** Corresponds to Fig 8B for diffusion coefficient $D = 0.048$ mm$^2$ms$^{-1}$.
(MP4)

**S3 Video. Animation of the excitation sequence in the SAN model structure (type 1).** Evolution of transmembrane potential in 3D corresponding to Fig 8A.
(MP4)

**S4 Video. Animation of the transmembrane potential $u$ in the SAN model structure (type 2).** Corresponds to $D_A = 0.160$ mm$^2$ms$^{-1}$, $D_S = 0.060$ mm$^2$ms$^{-1}$.
(MP4)

**S5 Video. Animation of the transmembrane potential *u* in the SAN model structure (type 2).** Corresponds to $D_A = 0.160$ mm$^2$ms$^{-1}$, $D_S = 0.052$ mm$^2$ms$^{-1}$.
(MP4)

**S6 Video. Animation of the excitation sequence in the SAN model structure (type 2).** Evolution of transmembrane potential in 3D corresponding to Fig 8D.
(MP4)

**S7 Video. Animation of the transmembrane potential $u^I$ in the intestine model.** Formation of plateaus with entrained frequencies ($t = 1$—3600 s).
(MP4)

**S8 Video. Animation of the transmembrane potential $u^I$ in the intestine model.** Onset and evolution of intestinal dysrhythmia pattern due to temporary conduction block ($t = 8000$—11600 s).
(MP4)

## Author Contributions

**Conceptualization:** Maxim Ryzhii, Elena Ryzhii.

**Data curation:** Maxim Ryzhii.

**Formal analysis:** Maxim Ryzhii, Elena Ryzhii.

**Funding acquisition:** Maxim Ryzhii.

**Investigation:** Maxim Ryzhii.

**Methodology:** Maxim Ryzhii, Elena Ryzhii.

**Project administration:** Maxim Ryzhii.

**Software:** Maxim Ryzhii.

**Supervision:** Maxim Ryzhii.

**Validation:** Maxim Ryzhii, Elena Ryzhii.

**Visualization:** Maxim Ryzhii.

**Writing – original draft:** Maxim Ryzhii, Elena Ryzhii.

**Writing – review & editing:** Maxim Ryzhii, Elena Ryzhii.

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
