## [Decision Letter · Decision Letter 0]

21 Oct 2021

PONE-D-21-29390Pacemaking function of two simplified cell modelsPLOS ONE

Dear Dr. Ryzhii,

Thank you for submitting your manuscript to PLOS ONE. After careful consideration, we feel that it has merit but does not fully meet PLOS ONE’s publication criteria as it currently stands. Therefore, we invite you to submit a revised version of the manuscript that addresses the points raised during the review process.

There are several issues that have been pointed out by reviewers and need to be considered point by point.

We look forward to receiving your revised manuscript.

Kind regards,

Agustín Guerrero-Hernandez

Academic Editor

PLOS ONE

Journal Requirements:

Reviewers' comments:

Reviewer's Responses to Questions

**Comments to the Author**

1. Is the manuscript technically sound, and do the data support the conclusions?

Reviewer #1: Yes

Reviewer #2: Yes

Reviewer #3: No

Reviewer #4: Yes

2. Has the statistical analysis been performed appropriately and rigorously? 

Reviewer #1: N/A

Reviewer #2: N/A

Reviewer #3: N/A

Reviewer #4: No

3. Have the authors made all data underlying the findings in their manuscript fully available?

Reviewer #1: No

Reviewer #2: Yes

Reviewer #3: Yes

Reviewer #4: Yes

4. Is the manuscript presented in an intelligible fashion and written in standard English?

Reviewer #1: Yes

Reviewer #2: Yes

Reviewer #3: Yes

Reviewer #4: Yes

5. Review Comments to the Author

Reviewer #1: This paper extends two simple variants of the Aliev-Panfilov and Corrado models in cardiac electrophysiology.

The authors then run a number of different simulations - either at the ODE level or in some simple tissue models - and report on the dynamical behaviours.

While there is some merit in this paper, there is very little that is new from a modelling perspective, a calibration perspective or in terms of the numerics. Basically, the authors just present a set of simulation results. For these reasons I cannot recommend it be published.

There are a number of other issues.

1. The introduction is poor. The authors are not clear on the issues around the use of different types of electrophysiology models. What are the issues exactly? If models are more complex, is there a calibration issue or a computational complexity issue. We can certainly run complex ion channel models in tissue. Line 17 is very questionable. This would need much more clarity than the rather crude discussion presented here.

2 The conclusions given in the last paragraph about the utility of these models in patient specific settings is not well made. There is no natural link between the simple hyper-parameters and patient specific data. This would need much more justification.

3. The modifications to existing models are very minor (eg the use of a sigmoid function). The models are similar to [43].

4. There is nothing new with respect to numerical approaches.

Reviewer #2: This paper reports on modifications to two existing simplified models of cellular electrophysiology. The authors have modified the Aliev-Panfilov (AP) and Corrado-Niederer (CN) models of excitable cells to yield pacemaker models capable of automaticity. As the authors have noted in their introduction, this approach has been previously applied to, e.g., the FitzHugh-Nagumo and original Hodgkin-Huxley models. The finding that a model of an excitable cell (with stable resting potential) can be converted to a pacemaking one is thus not in itself a new result. However, I agree with the authors that there are important applications for simplified (but well characterized) models of pacemaking cells.

The authors provide characterizations of how key properties, such as the frequency of pacemaking, depend on the parameters of the modified models. This analysis takes advantage of the simplified nature of the models and explains the behaviour in terms of phase portraits using standard methods for analysis of nonlinear systems. The authors then continue to use the modified models, along with the original formulations for excitable cells, to demonstrate the ability to qualitatively reproduce some basic behaviour when pacemaking cells are connected to excitable cells in physiological scenarios in the heart and intestine.

The work is well documented and clearly presented, and the methodology used is generally appropriate and correct. However, the relatively basic results presented here do not in my opinion go very far to demonstrate that the models can be applied for the authors' stated purposes (personalized medicine, medical device testing). While it is encouraging to see that the pacemaking models are able to "drive" excitable tissue in both scenarios tested, this defines a very narrow scope of baseline, normal physiological behaviour only. It is not clear how widely applicable the proposed models are.

In my view, this paper would be strengthened if the authors could explore the ability of the simplified models to reproduce more detailed/subtle properties of pacemaking. For example, can the modified AP model reproduce findings such as the the role of "discrete exit pathways" on micro-reentry and shift of the leading pacemaker site in the SA node, similar to the results presented by Karche et al. (the authors' reference 49, which employs variants of the Fenton-Karma model)? This would help to establish boundaries of applicability for the modified AP model and also contribute some additional understanding of whether or not the results of Karche et al are independent of the specifics of the simplified cellular model. I am less familiar with the literature on pacemaking in the intestine, but would assume that more detailed test cases could be identified in that application area as well.

Reviewer #3: This article develops two pacemaker models based on alterations to existing

phenomenological ionic models. The models are then applied to several example

systems to demonstrate their utility. In general, the authors do not

convincingly demonstrate the superiority of their new models, nor are the

examples physiologically meaningful. The paper also has many instances of

awkward language that need correction. Details are provided below.

The authors overestimate the cases where the simplified models are needed.

Given increases in computational power and numerical techniques, hundreds of

thousands of cells are easily handled on a desktop multicore machine. However,

this requires more detailed knowledge of computational techniques. The reviewer

agrees that there may be cases where a simple model may suffice.

The authors should demonstrate that their models are superior in some ways. For

example, are the waveforms more realistic? What parameters in the models control

behaviour (AP shape, frequency, resting level, etc.) and what are the limits? It is

not clear how cardiac models can be used for all tissue.

What situation are the authors modelling with the cardiac strands? Why is there

only one pacemaker? What do the waveforms look like? Electrotonic coupling is

important and depends on wave morphology.

The authors need to specify the species of the SAN they are trying to model. In

larger mammals, the SAN is isolated except at several discrete coupling points.

The gradients are missing in the example and are vital for function. See Munoz

Am J Physiol Heart Circ Physiol 2011. The authors need to show that they can be

incorporated and produce the correct behavior.

For the gastrointestinal example, again, somewhat realistic waveforms need to

be shown. How did the authors adjust the model? ICC slow waves last several

seconds which is much longer than anything shown. Also, there is a decreasing

frequency gradient along the intestine with sections of entrainment. Like the

other examples, this shows that the oscillators can be assembled and will show

activity but does not convincingly demonstrate that the essential elements of

the system under study can be recapitulated.

Reviewer #4: The paper considers variants of the Aliev-Panfilov and Corrado two-variable models to investigate nonlinear dynamic features of both isolated cells and 1D coupled pacemaker-excitable systems. As application examples of combined pacemaker-excitable systems, numerical simulations of 2D sinoatrial node and 3D intestine tissue were presented. Although the paper is well-written, I have major concerns about the numerical methods as described below:

1) Although the paper discusses the models parameters very well, less is said about the numerical methods. It is well known that all the electrocardiology models require accurate and precise numerical methods. In fact, the mesh size can greatly affect the wave velocity and the position of the depolarization and repolarization front. The type of space and time discretizations may affect the spiral and scroll waves dynamics. More discussions about these computational difficulties can be found in, for instance, [1-6]. Therefore, a major concern about the paper is that the numerical methods used for the simulations are not described. In addition, without showing the accuracy of the numerical methods, the results may not be reliable.

2) The accuracy of the method employed is discussed in the Numerical method Section. However, this is done only for single cell simulations. These results should be discussed in at least the 1D case and compare the results with an order-two approximations for both space and time.

3) The manuscript does not discuss the space discretization and the order of the approximation used.

4) As the manuscript considers only an explicit method for the time discretization, the standard stability criterion has to be forced. This may cause computational issues, especially in the 3D case. Would you please comment on the time step used for the 2D and 3D cases? How can one ensure that the results obtained are accurate and the numerical method employed did not affect the main paper's findings?

References:

[1] Efficiency of Semi-Implicit Alternating Direction Implicit Methods for Solving Cardiac Monodomain Model. Computers in Biology and Medicine. DOI: https://doi.org/10.1016/j.compbiomed.2020.104187

[2] High-order finite element methods for cardiac monodomain simulations. Front. Physiol. doi: 10.3389/fphys.2015.00217

[3] Parallel anisotropic mesh adaptivity with dynamic load balancing for cardiac electrophysiology, Journal of Computational Science, doi: https://doi.org/10.1016/j.jocs.2011.11.002

[4] Adaptive finite element simulation of ventricular fibrillation dynamics, Comput. Visual Sci. DOI: https://doi.org/10.1007/s00791-008-0088-y

[5] Simulation of cardiac electrophysiology on next-generation high-performance computers, DOI: https://doi.org/10.1098/rsta.2008.0298

[6] A Time-Dependent Adaptive Remeshing for Electrical Waves of the Heart.IEEE Transactions on Biomedical Engineering. DOI:10.1109/TBME.2007.905415

6. PLOS authors have the option to publish the peer review history of their article (what does this mean?). If published, this will include your full peer review and any attached files.

Reviewer #1: No

Reviewer #2: No

Reviewer #3: No

Reviewer #4: No

---

## [Author Response · Author response to Decision Letter 0]

5 Jan 2022

Response to Reviewer 1:

We thank Reviewer 1 for their critical comments which helped us to improve the paper. We have made corrections to the manuscript according to the Reviewer’s comments. The following provides our response to each comment in the order that it appeared in the report.

"1. The introduction is poor. The authors are not clear on the issues around the use of different types of electrophysiology models. What are the issues exactly? If models are more complex, is there a calibration issue or a computational complexity issue. We can certainly run complex ion channel models in tissue. Line 17 is very questionable. This would need much more clarity than the rather crude discussion presented here."

We have changed the text around Line 17-19 and added References 5-7, 20, 24 to support our statements. 

"2 The conclusions given in the last paragraph about the utility of these models in patient specific settings is not well made. There is no natural link between the simple hyper‐parameters and patient specific data. This would need much more justification."

To clarify justification for the utility of the considered models in the patient-specific settings, we have changed significantly “General remarks” subsection with the inclusion of corresponding references (19 and 34). Also, “Conclusion” has been modified (Lines 469-471, 474-475).

"3. The modifications to existing models are very minor (eg the use of a sigmoid function). The models are similar to [43]."

We cannot agree with the Reviewer’s statement due to the following points.

- The work [50] (former Ref. 43) refers to the Aliev-Panfilov model: ‘However, this model … does not encompass as many aspects of heart dynamics as do the ionic models (e.g. pacemaker activity).” p.5186, and in Table 1 (p. 5188) the pacemaking variant of the AP is absent.

- Corrado-Niederer model [19] represents a significant modification of the Mitchell-Schaeffer model [18] by changing the form of polynomial in the differential equation for transmembrane potential. This modification drastically changed the dynamics of the Mitchell-Schaeffer model and introduced robustness to unexpected and uncontrolled pacemaker behavior of the latter. The controlled pacemaker function of the Corrado-Niederer model clearly differs from that considered in [50].

- The sigmoid function itself was introduced in the classical Hodgkin-Huxley model (1952). In contrast to [50], where the role of the sigmoid function is unclear, we considered the influence of the function on the pCN model dynamics (Fig 5B).

- In contrast to [50], we considered several nonlinear characteristics (Figs 4 and 5) omitted in [50]. 

- In contrast to [50], we considered the dynamics of 1D coupled pAP-AP and pCN-CN systems (Figs 6 and 7).

- In contrast to [50], we considered the use of the pAP-AP and pCN-CN systems in the 2D SAN and 3D intestine tissue models (Figs 8 and 9).

"4. There is nothing new with respect to numerical approaches."

We used only standard numerical methods such as explicit Forward Euler (FE) and implicit Backward Euler (BE), as the methods are suitable for the solution of the ODE systems considered. The BE method is relatively simple to implement and is known to be absolutely stable, making it suitable for the solution of stiff differential equations (K. Atkinson, W. Han, D. Stewart, "Numerical solution of ordinary differential equations", John Wiley & Sons, Inc., 2008.; John C. Butcher, "Numerical methods for ordinary differential equations", Wiley, 2003). The BE method, for example, was used for the simulations with the original Corrado model [19]. We used the explicit FE method for the preliminary simulations and the BE method for the final results.

Response to Reviewer 2:

We thank Reviewer 2 for their useful and insightful comments, which helped us improve the manuscript. These comments are all implemented in the revised manuscript. The following provides our response to each comment in the order that it appeared in their report.

"The work is well documented and clearly presented, and the methodology used is generally appropriate and correct. However, the relatively basic results presented here do not in my opinion go very far to demonstrate that the models can be applied for the authors' stated purposes (personalized medicine, medical device testing). While it is encouraging to see that the pacemaking models are able to "drive" excitable tissue in both scenarios tested, this defines a very narrow scope of baseline, normal physiological behaviour only. It is not clear how widely applicable the proposed models are.

In my view, this paper would be strengthened if the authors could explore the ability of the simplified models to reproduce more detailed/subtle properties of pacemaking. For example, can the modified AP model reproduce findings such as the the role of "discrete exit pathways" on micro‐reentry and shift of the leading pacemaker site in the SA node, similar to the results presented by Karche et al. (the authors' reference 49, which employs variants of the Fenton‐Karma model)? This would help to establish boundaries of applicability for the modified AP model and also contribute some additional understanding of whether or not the results of Karche et al are independent of the specifics of the simplified cellular model."

Indeed, recent experimental findings demonstrate a more complex structure of the SAN, including insulating borders and exit pathways. To show the capabilities of the pAP model, we have changed the simulation setup and considered two different SAN structure types – with and without (original in the previous version of the manuscript) insulating borders with exit pathways. 

For this purpose, in “Methods - 2D SAN model” subsection:

- We have added panel B in Fig 3 (SAN structure with the insulating borders and exit pathways).

- We have rewritten the subsection completely starting from the 3rd paragraph to describe the modified simulation setup.

- We have added new results for SAN structure with the borders and exit pathways in Fig 8 (panels D-F). 

- For both types of the SAN structure, activation maps have been added (Fig 8C and Fig 8F).

- The description of the simulation results in “Results – 2D simulation of SAN” subsection has been changed.

- Table 1 has been split into two tables. The new Table 2 presents the parameters for the pAP and AP models used in the modified SAN simulation setups.

- We have added the references to Fedorov et al. 2012 [58], Kharche et al. 2017 [59], Li et al. 2014 [28], and Zyanterekov et al. 2019 [60].

- New supplementary videos have been added (S1 – S6 Videos).

 "I am less familiar with the literature on pacemaking in the intestine, but would assume that more detailed test cases could be identified in that application area as well."

To make the intestine model more realistic, we have made the following changes in “Methods - 3D intestine model” and “Results - 3D intestine model” subsections:

- Starting from the 3rd paragraph, the simulation setup description has been rewritten to describe the model changes. The mean circumference has been set to 44 mm, which corresponds to average-size animals like dogs or rabbits, and the simulation domain for both layers has been increased to 176x4800.

- In the former subsection, we have added references on recent experimental and modeling studies - Du et al. 2015 [61], Du et al. 2017 [62], Kararli 1995 [63], Angeli et al. 2013 [64].

- The pCN model parameters and the diffusion coefficients have been adjusted to demonstrate the formation of sections (plateaus) of frequency entrainment.

- The distribution of intrinsic frequencies along the y axis (Equation 13) has been changed.

- The parameters of the induced conduction block have been changed.

- Figure 9 demonstrates now not only updated snapshots of spatial distributions of the transmembrane potential but also the newly obtained distributions of intrinsic and entrained frequencies (Fig 9B), and both ICC and SMC action potentials at different time moments and positions in space (Figs 9C-9F). 

- The “Results - 3D simulation of intestine” subsection has been rewritten according to the newly obtained results. References to recent modeling and simulation studies have been added – Du et al. 2017 [62] and Parsons et al. 2015 [69].

- Two new supplementary videos have been added (S7 and S8 Videos).

Response to Reviewer 3:

We thank Reviewer 3 for their useful and critical comments, which helped us improve the manuscript. These comments are all implemented in the revised manuscript. The following provides our response to each comment in the order that it appeared in their report.

"This article develops two pacemaker models based on alterations to existing phenomenological ionic models. The models are then applied to several example systems to demonstrate their utility. In general, the authors do not convincingly demonstrate the superiority of their new models, nor are the examples physiologically meaningful. 

The paper also has many instances of awkward language that need correction. Details are provided below. "

We must admit that multiple instances of awkward language were present in the previous version. 

The language issues have been corrected.

"The authors overestimate the cases where the simplified models are needed. Given increases in computational power and numerical techniques, hundreds of thousands of cells are easily handled on a desktop multicore machine. However, this requires more detailed knowledge of computational techniques. The reviewer agrees that there may be cases where a simple model may suffice. "

We have modified the end of the first paragraph of “Introduction” and provided additional references [5-7, 20, 24] supporting the need for the simplified phenomenological models (Lines 17-19, 28-30), and a review [20].

"The authors should demonstrate that their models are superior in some ways. For example, are the waveforms more realistic? What parameters in the models control behaviour (AP shape, frequency, resting level, etc.) and what are the limits? It is not clear how cardiac models can be used for all tissue. "

The Aliev-Panfilov model is purely phenomenological, and the Corrado model is a reduced ionic model which can be considered phenomenological as well. As we mentioned in “Introduction” section, the advantages of these (and others like the FitzHugh-Nagumo model) are their simplicity (just two variables), allowing easy bifurcation analysis, the uniform description for both excitable and oscillating equation systems, and low computational cost – essential for real-time and interactive applications. The considered models have a well-defined excitation threshold (in contrast to the FHN model) and convenient intrinsic frequency control in a wide range (Figs 4 and 5). Another use of the models can be preliminary simulation setups for hypothesis testing and verification, as well as model design and tuning before implementing complex and computationally expensive ion-channel models. This approach was used in the paper by Teplenin et al. Phys. Rev. X 8, 021077, 2018 [32]. We have also modified “General remarks” section.

"What situation are the authors modelling with the cardiac strands? Why is there only one pacemaker?"

The strands of tissue (not exactly cardiac) were considered to evaluate and compare the pacemaking properties of both pAP and pCN models with varying coupling between cells (coupling load). As Fig 6 shows, the models demonstrate rather different dynamic behavior. 

In this modeling setup (section “Coupled 1D pacemaker-excitable systems”) we evaluated the ability of a single pacemaker cell to drive relatively long strands of the excitable cells. It is essential for comprehensive abstracted heart models with real-time simulation capabilities, where a single pacemaker cell represents a group of ion-channel pacemaker model cells [43]. We have modified the section accordingly (Lines 165-167).

"What do the waveforms look like? Electrotonic coupling is important and depends on wave morphology."

We have added Figs 7C and 7D with temporal evolution of action potential waveforms for both pAP-AP and pCN-CN 1D coupled systems at different coupling coefficients. The corresponding text has been added at the end of “Results – Synchronization behavior…” subsection.

"The authors need to specify the species of the SAN they are trying to model."

We have added the citation to an experimental paper by Opthof, Cardiovasc Drugs and Ther. 1988 [54] and noted the canine heart in the second paragraph of “Methods - 2D SAN model” subsection.

"In larger mammals, the SAN is isolated except at several discrete coupling points. The gradients are missing in the example and are vital for function. See Munoz Am J Physiol Heart Circ Physiol 2011. The authors need to show that they can be incorporated and produce the correct behavior."

We agree with the reviewer. 

Indeed, recent experimental findings demonstrate a more complex structure of the SAN, including insulating borders and exit pathways. To support the capabilities of the pAP model, we have changed the simulation setup and considered two different SAN structure types – with and without (original in the previous version of the manuscript) insulating borders with exit pathways. 

For this purpose, in “Methods - 2D SAN model” subsection:

- We have added panel B in Fig 3 (SAN structure with the insulating borders and exit pathways).

- We have rewritten the subsection completely, starting from the 3rd paragraph to describe the changed simulation setup.

- We have added new results for SAN structure with the borders and exit pathways in Fig 8 (panels D-F). 

- For both types of the SAN structure, activation maps have been added (Fig 8C and Fig 8F).

- We have changed the description of the simulation results in “Results – 2D simulation of SAN” subsection.

- We have added the references to Fedorov et al. 2012 [58], Kharche et al. 2017 [59], Li et al. 2014 [28], and Zyanterekov et al. 2019 [60].

- New supplementary videos have been added (S1 – S6 Videos).

"For the gastrointestinal example, again, somewhat realistic waveforms need to be shown. How did the authors adjust the model? ICC slow waves last several seconds which is much longer than anything shown. Also, there is a decreasing frequency gradient along the intestine with sections of entrainment. Like the other examples, this shows that the oscillators can be assembled and will show activity but does not convincingly demonstrate that the essential elements of the system under study can be recapitulated."

To make the intestine model more realistic, we have made the following changes in “Methods - 3D intestine model” subsection:

- Starting from the 3rd paragraph, the simulation setup description has been rewritten to describe the model changes. In particular, the mean circumference has been set to 44 mm, which corresponds to average-size animals like dogs or rabbits.

- In the subsection, we have added references on recent experimental and modeling studies - Du et al. 2015 [61], Du et al. 2017 [62], Kararli 1995 [63], Angeli et al. 2013 [64].

- The pCN model parameters and the diffusion coefficients have been adjusted to demonstrate the formation of sections (plateaus) of frequency entrainment.

- The distribution of intrinsic frequencies along the y axis (Equation 13) has been modified.

- The parameters of the induced conduction block have been modified.

Correspondingly, the “Results - 3D simulation of intestine” subsection has been updated:

- Figure 9 demonstrates now not only updated snapshots of spatial distributions of the transmembrane potential but also the newly obtained distributions of intrinsic and entrained frequencies (Fig 9B) and both ICC and SMC action potentials at different time moments and positions in space (Figs 9C-9F). 

- The subsection has been rewritten according to the newly obtained results. References to recent modeling and simulation studies have been added – Du et al. 2017 [62] and Parsons et al. 2015 [69].

- Two new supplementary videos have been added (S7 and S8 Videos).

Response to Reviewer 4:

We thank Reviewer 4 for their useful and insightful comments, which helped us improve the manuscript. These comments are all implemented in the revised manuscript. The following provides our response to each comment in the order that it appeared in their report.

"Although the paper is well‐written, I have major concerns about the numerical methods as described below: 

1) Although the paper discusses the models parameters very well, less is said about the numerical methods. It is well known that all the electrocardiology models require accurate and precise numerical methods. In fact, the mesh size can greatly affect the wave velocity and the position of the depolarization and repolarization front. The type of space and time discretizations may affect the spiral and scroll waves dynamics. More discussions about these computational difficulties can be found in, for instance, [1‐6]. Therefore, a major concern about the paper is that the numerical methods used for the simulations are not described. In addition, without showing the accuracy of the numerical methods, the results may not be reliable."

Both explicit Forward Euler (FE) and implicit Backward Euler (BE) methods have order one accuracy. The BE method is relatively simple to implement and is known to be absolutely stable, making it suitable for the solution of stiff differential equations (K. Atkinson, W. Han, D. Stewart, "Numerical solution of ordinary differential equations", John Wiley & Sons, Inc., 2008.; John C. Butcher, "Numerical methods for ordinary differential equations", Wiley, 2003). So, we used the explicit FE method for the preliminary simulations and the BE method for the final results. 

In the 1D simulations, we used a coupling coefficient d = D/Δx2, not dependent on spatial step size. Time steps for 1D pAP-AP and pCN-CN coupled systems are presented in Tables 1 and 3.

“Numerical methods” subsection has been modified. We have split Table 1 into two tables. New Tables 2 and 3 demonstrate, along with other parameters, time and space steps for the 2D SAN and 3D intestine models.

"2) The accuracy of the method employed is discussed in the Numerical method Section. However, this is done only for single cell simulations. These results should be discussed in at least the 1D case and compare the results with an order‐two approximations for both space and time.

3) The manuscript does not discuss the space discretization and the order of the approximation used."

We agree with the Reviewer that implementing other more precise and complex methods may be beneficial for such simulations in the case of a research study of the electrophysiology of the SAN and intestine. The main aim of our paper is the consideration of the pacemaking functions of the AP and CN models and examples of their application (to demonstrate the ability of the considered models describing pacemaking tissue behavior in the SAN and intestine). Thus, we limited ourselves to implementing FE and implicit BE methods only. Even the explicit FE method with proper time step is considered accurate enough for computationally efficient models (namely, modified Aliev-Panfilov model), see, for example, “Computationally efficient model of myocardial electromechanics for multiscale simulations” by F. Syomin et al. PLOS One, (2021) (https://doi.org/10.1371/journal.pone.0255027). Implementation of both FE and BE methods Cardiac CHASTE software was demonstrated in the works “CHASTE: incorporating a novel multi-scale spatial and temporal algorithm into a large-scale open source library” by M.O. Bernabeu, Phil. Trans. R. Soc. A (2009) (https://doi.org/10.1098/rsta.2008.0309 ) and “Cellular cardiac electrophysiology modeling with Chaste and CellML”, by J. Cooper, Front. Physiol. (2015) (https://doi.org/10.3389/fphys.2014.00511 ).

A comparison of the accuracy of FE results relative to that obtained with BE method for 0D simulations is presented in Table 4. For 1D coupled pacemaker-excitable systems, the maximum relative frequency error with dt = 0.1 ms was also about 0.16%. We have also added Supplement S1 Fig with the comparison of the accuracy of FE and BE methods used in the simulations. Pertinent MATLAB and CellML codes have been placed at GitHub public repository: https://github.com/mryzhii/Simplified-pacemaker-cell-models .

"4) As the manuscript considers only an explicit method for the time discretization, the standard stability criterion has to be forced. This may cause computational issues, especially in the 3D case. Would you please comment on the time step used for the 2D and 3D cases? How can one ensure that the results obtained are accurate and the numerical method employed did not affect the main paper's findings?"

We used both the explicit Forward Euler method for the preliminary simulations and the implicit Backward Euler method for final results. For the 2D and 3D simulations, the results obtained with both methods are visually the same, and further reduction of time and space steps did not lead to any visual difference. In the BE method, the absolute tolerance (for each cell in 1D, 2D, and 3D cases) was set to 10^(-7) with the maximum number of iterations in the inner loop 20. The latter was not exceeded in all simulations (Lines 263-267).

For the intestine simulations, the time and spatial steps were smaller than that used in a similar study of gastric electro-mechanical activity (dt=2.5 ms and dx=0.25 mm vs. dt=100 ms, dx=0.5 mm in Brandstaeter et al. 2018 [39]), considering a fivefold increase in the frequency in our case (Table 3).

---

## [Decision Letter · Decision Letter 1]

16 Mar 2022

PONE-D-21-29390R1Pacemaking function of two simplified cell modelsPLOS ONE

Dear Dr. Ryzhii,

Thank you for submitting your manuscript to PLOS ONE. After careful consideration, we feel that it has merit but does not fully meet PLOS ONE’s publication criteria as it currently stands. Therefore, we invite you to submit a revised version of the manuscript that addresses the points raised during the review process. Specifically: Please, include the limitations of this type of model in the discussion.

We look forward to receiving your revised manuscript.

Kind regards,

Agustín Guerrero-Hernandez

Academic Editor

PLOS ONE

Journal Requirements:

Reviewers' comments:

Reviewer's Responses to Questions

**Comments to the Author**

1. If the authors have adequately addressed your comments raised in a previous round of review and you feel that this manuscript is now acceptable for publication, you may indicate that here to bypass the “Comments to the Author” section, enter your conflict of interest statement in the “Confidential to Editor” section, and submit your "Accept" recommendation.

Reviewer #2: All comments have been addressed

Reviewer #3: (No Response)

Reviewer #4: (No Response)

2. Is the manuscript technically sound, and do the data support the conclusions?

Reviewer #2: (No Response)

Reviewer #3: Yes

Reviewer #4: Yes

3. Has the statistical analysis been performed appropriately and rigorously? 

Reviewer #2: (No Response)

Reviewer #3: N/A

Reviewer #4: N/A

4. Have the authors made all data underlying the findings in their manuscript fully available?

Reviewer #2: (No Response)

Reviewer #3: Yes

Reviewer #4: Yes

5. Is the manuscript presented in an intelligible fashion and written in standard English?

Reviewer #2: (No Response)

Reviewer #3: Yes

Reviewer #4: Yes

6. Review Comments to the Author

Reviewer #2: (No Response)

Reviewer #3: In this version of the paper, the authors have tried to make the examples more

realistic and have, for the most part succeeded. However, their type 2 SAN

simulations look odd. That put a passive barrier around their SAN instead of an

insulating one. As such, electrotonic interactions occur across the barrier

which affects propagation in the vicinity. This should not occur.

The authors should list the limitations of their model. While such a model can

be useful at times, it is also important to say when it cannot. For example,

how limited is the morphology? Can it support a bursting mode on top of the

plateau? Since voltage is normalized, it appears that resting level differences

cannot be incorporated.

Reviewer #4: The authors have addressed all my concerns with the first version of their manuscript. After the approval of the editor, I think the revised manuscript can be accepted for publication in the PLOS One Journal.

7. PLOS authors have the option to publish the peer review history of their article (what does this mean?). If published, this will include your full peer review and any attached files.

Reviewer #2: No

Reviewer #3: No

Reviewer #4: No

---

## [Author Response · Author response to Decision Letter 1]

22 Mar 2022

Response to Reviewer 3:

We thank Reviewer 3 for the comments, which helped us improve the manuscript. These comments are implemented in the revised manuscript. The following provides our response to the comments.

We have renamed the Section “General remarks” to “Limitations and general remarks”, added the text below (lines 444-474), and added Supplement S2 Fig describing methods for shifting of the resting (MDP) and amplitude/peak (POP) levels.

“…The results in Figs 8 and 9 are presented for illustration purposes and not aimed at a detailed study of these organs. In particular, in the SAN type 2 model (Fig 3B), the tissue of the borders surrounding the pacemaking area is of passive type, blocking the spread of excitation [28]. Such tissue, being not exactly of an electrically insulating type [59, 60], causes electrotonic interactions with the surrounding SAN and atrial tissues. Better selection of the boundary tissue properties can be beneficial for more realistic simulations of the SAN and its exit pathway functions. Moreover, additional tuning of the parameters to clinical data might be necessary for accurate simulations, in particular, to construct patient-specific models. This can be realized by applying, for example, a robust and clinically tractable protocol and fitting algorithm [19] for characterizing cardiac electrophysiology properties by simplest two-variable cell models, such as the pAP-AP and pCN-CN coupled systems. 

For the simulations where the differences of resting and/or peak levels are necessary, the MDP and POP values of the considered models can be modified. The addition of a constant to the term u and the replacement of unity in the term (1 −u) allow shifting of the MDP and POP levels, respectively, though the exact resulting values of the latters cannot be determined directly (see Supplement S2 Fig for the details). Also, such modification may require an adjustment of the intrinsic frequencies (for example, with the parameters bAP and bCN). The above modifications of the MDP and POP together with modulation of the intrinsic frequency with the parameters b may allow simulation of a kind of tonic bursting [29] - a firing behavior in which a neuron cell fires a certain number of spikes on the top of the plateau and is silent for a certain amount of time. Though, compared to the variety of specific neuronal cell models [29], the pAP and pCN models may not be the best choice for modeling of neuronal systems. 

Another well-known disadvantage of the phenomenological models like pAP-AP and pCN-CN is their limited ability to represent action potential morphology under varying physiological conditions, e.g., the effect of variation of concentration of particular ions, which is necessary for simulations of complex cardiac diseases such as ion channelopathies. However, …”

Response to Reviewer 4:

We thank Reviewer 4 for useful and insightful comments, which helped us improve the manuscript.

---

## [Decision Letter · Decision Letter 2]

30 Mar 2022

Pacemaking function of two simplified cell models

PONE-D-21-29390R2

Dear Dr. Ryzhii,

We’re pleased to inform you that your manuscript has been judged scientifically suitable for publication and will be formally accepted for publication once it meets all outstanding technical requirements.

Kind regards,

Agustín Guerrero-Hernandez

Academic Editor

PLOS ONE

Additional Editor Comments (optional):

Reviewers' comments:

Reviewer's Responses to Questions

**Comments to the Author**

1. If the authors have adequately addressed your comments raised in a previous round of review and you feel that this manuscript is now acceptable for publication, you may indicate that here to bypass the “Comments to the Author” section, enter your conflict of interest statement in the “Confidential to Editor” section, and submit your "Accept" recommendation.

Reviewer #3: All comments have been addressed

2. Is the manuscript technically sound, and do the data support the conclusions?

Reviewer #3: (No Response)

3. Has the statistical analysis been performed appropriately and rigorously? 

Reviewer #3: (No Response)

4. Have the authors made all data underlying the findings in their manuscript fully available?

Reviewer #3: (No Response)

5. Is the manuscript presented in an intelligible fashion and written in standard English?

Reviewer #3: (No Response)

6. Review Comments to the Author

Reviewer #3: (No Response)

7. PLOS authors have the option to publish the peer review history of their article (what does this mean?). If published, this will include your full peer review and any attached files.

Reviewer #3: No

---

## [Editor Report · Acceptance letter]

1 Apr 2022

PONE-D-21-29390R2 

Pacemaking function of two simplified cell models 

Dear Dr. Ryzhii:

I'm pleased to inform you that your manuscript has been deemed suitable for publication in PLOS ONE. Congratulations! Your manuscript is now with our production department. 

Kind regards, 

on behalf of

Dr. Agustín Guerrero-Hernandez 

Academic Editor

PLOS ONE